# The NeuroML ecosystem for standardized multi-scale modeling in neuroscience

Ankur Sinha[1†], Padraig Gleeson[1*†], Bóris Marin[2], Salvador Dura-Bernal[3,4], Sotirios Panagiotou[5], Sharon Crook[6], Matteo Cantarelli[7], Robert C Cannon[8], Andrew P Davison[9], Harsha Gurnani[10], Robin Angus Silver[1*]

[1]Department of Neuroscience, Physiology and Pharmacology, University College London, London, United Kingdom; [2]Universidade Federal do ABC, São Bernardo do Campo, Brazil; [3]SUNY Downstate Medical Center, Brooklyn, United States; [4], Center for Biomedical Imaging and Neuromodulation, Nathan Kline Institute for Psychiatric Research, Orangeburg, United States; [5]Erasmus University Rotterdam, Rotterdam, Netherlands; [6]Arizona State University, Tempe, United States; [7]MetaCell Ltd, Cambridge, United States; [8]Opus2 International Ltd, London, United Kingdom; [9]CNRS, Gif-Sur-Yvette, France; [10]University of Washington, Seattle, United States

**\*For correspondence:**
p.gleeson@ucl.ac.uk (PG);
a.silver@ucl.ac.uk (RAS)

[†]These authors contributed equally to this work

## eLife Assessment

This **important** work presents a consolidated overview of the NeuroML2 open community standard and provides **convincing** evidence for its central role within a broader software ecosystem for the development of neuronal models that are open, shareable, reproducible, and interoperable. A major strength of the work is the continued development over more than two decades to establish, maintain, and adapt this standard to meet the evolving needs of the field. This work is of broad interest to the sub-cellular, cellular, computational, and systems neuroscience communities undertaking studies involving theory, modeling, and simulation.

**Abstract** Data-driven models of neurons and circuits are important for understanding how the properties of membrane conductances, synapses, dendrites, and the anatomical connectivity between neurons generate the complex dynamical behaviors of brain circuits in health and disease. However, the inherent complexity of these biological processes makes the construction and reuse of biologically detailed models challenging. A wide range of tools have been developed to aid their construction and simulation, but differences in design and internal representation act as technical barriers to those who wish to use data-driven models in their research workflows. NeuroML, a model description language for computational neuroscience, was developed to address this fragmentation in modeling tools. Since its inception, NeuroML has evolved into a mature community standard that encompasses a wide range of model types and approaches in computational neuroscience. It has enabled the development of a large ecosystem of interoperable open-source software tools for the creation, visualization, validation, and simulation of data-driven models. Here, we describe how the NeuroML ecosystem can be incorporated into research workflows to simplify the construction, testing, and analysis of standardized models of neural systems, and supports the FAIR (Findability, Accessibility, Interoperability, and Reusability) principles, thus promoting open, transparent and reproducible science.

## Introduction

Development of an in-depth, mechanistic understanding of brain function in health and disease requires different scientific approaches spanning multiple scales, from gene expression to behavior. Although 'wet' experimental approaches are essential for characterizing the properties of neural systems and testing hypotheses, theory and modeling are critical for exploring how these complex systems behave across a wider range of conditions, and for generating new experimentally testable, physically plausible hypotheses. Theory and modeling also provide a way to integrate a panoply of experimentally measured parameters, functional properties, and responses to perturbations into a physio-chemically coherent framework that reproduces the properties of the neural system of interest (*Einevoll et al., 2019*; *Yao et al., 2022*; *Poirazi and Papoutsi, 2020*; *Gurnani and Silver, 2021*; *Gleeson et al., 2018*; *Cayco-Gajic et al., 2017*; *Billings et al., 2014*; *Vervaeke et al., 2010*; *Kriener et al., 2022*; *Billeh et al., 2020*; *Markram et al., 2015*).

Computational models in neuroscience often focus on different levels of description. For example, a cellular physiologist may construct a complex multi-compartmental model to explain the dynamical behavior of an individual neuron in terms of its morphology, biophysical properties, and ionic conductances (*Hay et al., 2011*; *De Schutter and Bower, 1994*; *Migliore et al., 2005*). In contrast, to relate neural population activity to sensory processing and behavior, a systems neurophysiologist may build a circuit-level model consisting of thousands of much simpler integrate-and-fire neurons (*Lapicque, 1907*; *Potjans and Diesmann, 2014*; *Brunel, 2000*). Domain specific tools have been developed to aid the construction and simulation of models at varying levels of biological detail and scales. An ecosystem of diverse tools is powerful and flexible, but it also creates serious challenges for the research community (*Cannon et al., 2007*). Each tool typically has its own design, features, Application Programming Interface (API) and syntax, a custom set of utility libraries, and finally, a distinct machine-readable representation of the model's physiological components. This represents a complex landscape for users to navigate. Additionally, models developed in different simulators cannot be mixed and matched or easily compared, and the translation of a model from one tool-specific implementation to another can be non-trivial and error-prone. This fragmentation in modeling tools and approaches can act as a barrier to neuroscientists who wish to use models in their research, as well as impede how Findable, Accessible, Interoperable, and Reusable (FAIR) models are (*Wilkinson et al., 2016*).

To counter fragmentation and promote cooperation and interoperability within and across fields, standardization is required. The International Neuroinformatics Co-ordinating Facility (INCF) (*Abrams et al., 2022*) has highlighted the need for standards to 'make research outputs machine-readable and computable and are necessary for making research FAIR' (*INCF, 2023*). In biology, several community standards have been developed to describe experimental data (e.g. Brain Imaging Data Structure [BIDS; *Gorgolewski et al., 2016*], Neurodata Without Borders [NWB; *Teeters et al., 2015*]) and computational models (e.g. Systems Biology Markup Language [SBML; *Hucka et al., 2003*], CellML [*Lloyd et al., 2004*], Scalable Open Network Architecture TemplAte [SONATA; *Dai et al., 2020*], PyNN [*Davison et al., 2008*] and Neural Open Markup Language [NeuroML; *Gleeson et al., 2010*]). These standards have enabled open and interoperable ecosystems of software applications, libraries, and databases to emerge, facilitating the sharing of research outputs, an endeavor encouraged by a growing number of funding agencies and scientific journals.

The initial version of the NeuroML standard, version 1 (NeuroMLv1), was originally conceived as a model description format (*Goddard et al., 2001*) and implemented as a three-layered, declarative, modular, simulator-independent language (*Gleeson et al., 2010*). NeuroMLv1 could describe detailed neuronal morphologies and their biophysical properties as well as specific instantiations of networks. It enabled the archiving of models in a standardized format and addressed the issue of simulator fragmentation by acting as the common language for model exchange between established simulation environments—NEURON (*Hines and Carnevale, 1997*; *Awile et al., 2022*), GENESIS (*Bower and Beeman, 1998*), and MOOSE (*Ray and Bhalla, 2008*). While solving a number of long-standing problems in computational neuroscience, NeuroMLv1 had several key limitations. The most restrictive of these was that the dynamical behavior of model elements was not formally described in the standard itself, making it only partially machine readable. Information on the dynamics of elements (i.e. how the state variables should evolve in time) was only provided in the form of human-readable documentation, requiring the developers of each new simulator to re-implement the behavior of these elements

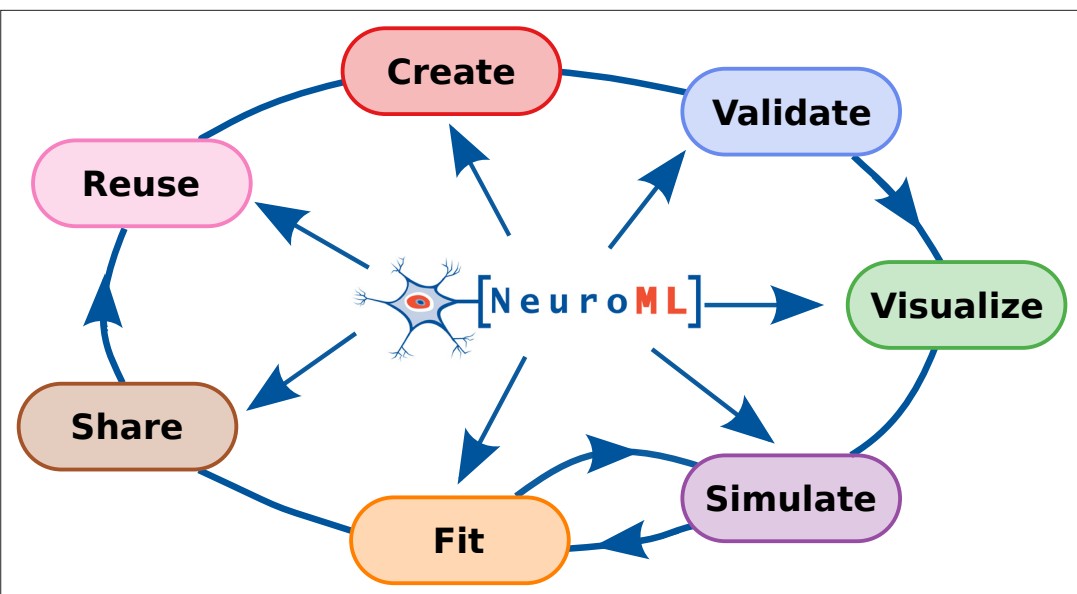

**Figure 1.** The NeuroML software ecosystem supports all stages of the model development life cycle.

in their native format. Additionally, the introduction of new model components required updates to the standard and all supporting simulators, making extension of the language difficult. Finally, the use of Extensible Markup Language (XML) as the primary interface language limited usability—applications would generally have to add their own code to read/write XML files.

To address these limitations, NeuroML was redesigned from the ground up in version 2 (NeuroMLv2) using the Low Entropy Modeling Specification (LEMS) language (*Cannon et al., 2014*). LEMS was designed to define a wide range of physio-chemical systems, enabling the creation of fully machine-readable, formal definitions of the structure and dynamics of any model components. Modeling elements in NeuroMLv2 (cells, ion channels, synapses) have their mathematical and structural definitions described in LEMS (e.g. the parameters required and how the state variables change with time). Thus, NeuroMLv2 retains all the features of NeuroMLv1—it remains modular, declarative, and continues to support multiple simulation engines—but unlike version 1, it is extensible, and all specifications are fully machine-readable. NeuroMLv2 also moved to Python as its main interface language and provides a comprehensive set of Python libraries to improve usability (*Vella et al., 2014*), with XML retained as a machine-readable serialization format (i.e. the form in which the model files are saved/shared).

Since its release in 2014, the NeuroMLv2 standard, the software ecosystem, and the community have all steadily grown. An open, community-based governance structure was put in place—an elected Editorial Board, overseen by an independent Scientific Committee, maintains the standard and core software tools—APIs, reference simulators, and utilities. Although these tools were initially focused on enabling the simulation of models on multiple platforms, they have been expanded to support all stages of the model life cycle (*Figure 1*). Modelers can use these tools to easily create, inspect and visualize, validate, simulate, fit and optimize, share and disseminate NeuroMLv2 models and outputs (*Billings et al., 2014*; *Cayco-Gajic et al., 2017*; *Gurnani and Silver, 2021*; *Kriener et al., 2022*; *Gleeson et al., 2019b*). To provide clear, concise, searchable information for both users and developers, the NeuroML documentation has been significantly expanded and re-deployed using the latest modern web technologies (https://docs.neuroml.org). Increased community-wide collaborations have also extended the software ecosystem well beyond the NeuroMLv2 tools developed by the NeuroML team: additional simulators such as Brian (*Stimberg et al., 2019*), NetPyNE (*Dura-Bernal et al., 2019*), Arbor (*Akar et al., 2019*) and EDEN (*Panagiotou et al., 2022*) all support NeuroMLv2. We have worked to ensure interoperability with other structured formats for model development in neuroscience such as PyNN (*Davison et al., 2008*) and SONATA (*Dai et al., 2020*). Platforms for collaboratively developing, visualizing, and sharing NeuroML models (Open Source Brain (OSB) *Gleeson et al., 2019b*) as well as a searchable database of NeuroML model components NeuroML

Database (NeuroML-DB) (*Birgiolas et al., 2023*) have been developed. These enhancements, driven by an ever-expanding community, have helped NeuroMLv2 grow into a standard that has been officially endorsed by international organizations such as the INCF and COmputational Modeling in Biology NEtwork (COMBINE) (*Hucka et al., 2015*), and that is now sufficiently mature to be incorporated into a wide range of research workflows.

In this paper, we provide an overview of the current scope of version 2 of the NeuroML standard, describe the current software ecosystem and community, and outline the extensive resources to assist researchers in incorporate NeuroML into their modeling work. We demonstrate, with examples, that NeuroML supports users at all stages of the model development life cycle (*Figure 1*) and promotes FAIR principles in computational neuroscience. We highlight the various NeuroML tools and libraries, additional utilities, supported simulation engines, and the related projects that build upon NeuroML for automated model validation, advanced analysis, visualization, and sharing/re-use of models. Finally, we summarize the organizational aspects of NeuroML, its governance structure and its community.

## Results
## NeuroML provides a ready-to-use set of curated model elements

A central aim of the NeuroML initiative is to enable and encourage the use of multi-scale biophysically detailed models of neurons and neuronal circuits in neuroscience research. The initiative takes a range of steps to achieve this aim.

NeuroML provides users with a curated library of model elements that form the NeuroML standard (An index of all the model elements included in version 2.3 of NeuroML, with links to further online documentation, is provided in *Tables 1 and 2*; *Figure 2*). The standard is maintained by the NeuroML Editorial Board that has identified a fundamental set of model types to support, to ensure that a significant proportion of commonly used neurobiological modeling entities can be described with the language. This includes (but is not limited to): active membrane conductances (using Hodgkin-Huxley style [*Hodgkin and Huxley, 1952*] or kinetic scheme-based ionic conductances), multiple synapse and plasticity mechanisms, detailed multi-compartmental neuron models with morphologies and biophysical properties, abstract point neuron models, and networks of such cells spatially arranged in populations, connected by targeted projections, receiving spiking and currently based inputs.

The NeuroMLv2 standard consists of two levels that are designed to enable users to easily create their models without worrying about simulator-specific details. The first level defines a formal 'schema' for the standard model elements, their attributes/parameters (e.g. an integrate and fire cell model and its necessary attributes: a threshold parameter, a reset parameter, etc.), and the relationships between them (e.g. a network contains populations; a multi-compartmental cell morphology contains segments). This allows the validation of the completeness of the description of individual NeuroML model elements and models, *prior to simulation*. The second level defines the underlying dynamical behavior of the model elements (e.g. how the time-varying membrane potential of a cell model is to be calculated). Most users do not need to interact with this level (which is enabled by LEMS), which, among other features, enables the automated translation of *simulator-independent* NeuroML models into *simulator-specific* code.

Thus, modelers can use the standard NeuroML elements to conveniently build simulator-independent models, while also being able to examine and extend the underlying implementations of models. As a simulator-independent language, NeuroML also promotes interoperability between different computational modeling tools, and as a result, the standard library is complemented by a large, well-maintained ecosystem of software tools that support all stages of the model life cycle—from creation, analysis, simulation, and fitting, to sharing and reuse. Finally, as discussed in later sections, for advanced use cases where the existing NeuroML model building blocks are insufficient, NeuroML also includes a framework for creating and including new model elements.

## NeuroML is a modular, structured language for defining FAIR models

NeuroMLv2 is a modular, structured, hierarchical, simulator-independent format. All NeuroML elements are formally defined, independent, and self-contained with hierarchical relationships between them. An 'ionic conductance' model element in NeuroML, for example, can contain zero, one, or more 'gates' and be added into a 'cell' model element along with a 'morphology' element, which can then

**Table 1.** Index of standard NeuroMLv2 ComponentTypes.

**Core components**

| | | |
|---|---|---|
| annotation | bqbiol_encodes | bqbiol_hasPart |
| bqbiol_hasProperty | bqbiol_hasTaxon | bqbiol_hasVersion |
| bqbiol_is | bqbiol_isDescribedBy | bqbiol_isEncodedBy |
| bqbiol_isHomologTo | bqbiol_isPartOf | bqbiol_isPropertyOf |
| bqbiol_isVersionOf | bqbiol_occursIn | bqmodel_is |
| bqmodel_isDerivedFrom | bqmodel_isDescribedBy | rdf_Bag |
| rdf_Description | rdf_li | rdf_RDF |
| property | point3DWithDiam | notes |

**Core dimensions**

| | | |
|---|---|---|
| area | capacitance | charge |
| charge_per_mole | concentration | conductance |
| conductance_per_voltage | conductanceDensity | current |
| currentDensity | idealGasConstantDims | length |
| per_time | per_voltage | permeability |
| resistance | resistivity | rho_factor |
| specificCapacitance | substance | temperature |
| time | voltage | volume |

**Abstract cell models**

| | | |
|---|---|---|
| adExIaFCell | fitzHughNagumoCell | hindmarshRose1984Cell |
| iafCell | iafRefCell | iafTauCell |
| iafTauRefCell | izhikevich2007Cell | izhikevichCell |
| pinskyRinzelCA3Cell | | |

**ComponentTypes related to biophysically detailed cells**

| | | |
|---|---|---|
| biophysical Properties | biophysicalProperties2CaPools | cell |
| cell2CaPools | concentration Model | decayingPoolConcentrationModel |
| distal | distalProperties | fixedFactorConcentrationModel |
| fixedFactorConcentrationModelTraub | from | include |
| inhomogeneousParameter | inhomogeneousValue | initMembPotential |
| intracellular Properties | intracellularProperties2CaPools | member |
| membraneProperties | membraneProperties2CaPools | morphology |
| parent | path | pointCellCondBased |
| pointCellCondBasedCa | proximal | proximalProperties |
| segment | segment Group | species |
| spikeThresh | subTree | to |
| variable Parameter | channel Density | channelDensityGHK |
| channelDensityGHK2 | channelDensityNernst | channelDensityNernstCa2 |
| channelDensityNonUniform | channelDensityNonUniformGHK | channelDensityNonUniformNernst |
| channelDensityVShift | channelPopulation | channelPopulationNernst |

**ComponentTypes related to ion channels**

| | | |
|---|---|---|
| fixedTimeCourse | forward Transition | gate |
| gateFractional | gateHHInstantaneous | gateHHrates |
| gateHHratesInf | gateHHratesTau | gateHHratesTauInf |

*Table 1 continued on next page*

*Table 1 continued*

**Core components**

| | | |
|---|---|---|
| gateHHtauInf | gateKS | HHExpLinearRate |
| HHExpLinearVariable | HHExpRate | HHExpVariable |
| HHSigmoidRate | HHSigmoidVariable | ionChannel |
| ionChannelHH | ionChannelKS | ionChannelPassive |
| ionChannelVShift | KSState | KSTransition |
| open State | q10ConductanceScaling | q10ExpTemp |
| q10Fixed | reverse Transition | sub Gate |
| tauInfTransition | vHalfTransition | closedState |

fit into a 'population' of a 'network' (**Figure 2**). To support the range of electrical properties found in biological neurons, ionic conductances with distinct ionic selectivities and dynamics can be generated in NeuroML through the inclusion of different types of gates (e.g. activation, inactivation), their dependence on variables such as voltage and [$Ca^{2+}$] and their reversal potential. Cell types with different functional and biophysical properties can then be generated by conferring combinations of ionic conductances on their membranes. The conductance density can be adjusted to generate the electrophysiological properties found in real neurons. In practice, many examples of ionic conductances that underlie the electrical behavior of neurons are already available in NeuroMLv2 and can simply be inserted into a cell membrane (**Figure 2**). Indeed, a model element, once defined in NeuroML, acts as a building block that may be reused any number of times within or across models. Elements such as ionic conductances, cell biophysics, cell morphologies, and cell definitions that incorporate them can be serialized in separate files and 'included' in other models (e.g. morphologies https://docs.neuroml. org/Userdocs/ImportingMorphologyFiles.html#neuroml2). Such reuse of model components speeds model construction and prototyping irrespective of the simulation engine used.

The defined structure of each model element and the relationships between them inform users of exactly how model elements are to be created and combined. This encourages the construction of well-structured models, reduces errors and redundancy, and ensures that FAIR principles are firmly embedded in NeuroML models and the ecosystem of tools. As we will see in the following sections, NeuroML's formal structure also enables features such as model validation prior to simulation, translation into simulation specific formats, and the use of NeuroML as a common language of exchange between different tools.

## NeuroML supports a large ecosystem of software tools that cover all stages of the model life cycle

Model building and the generation of scientific knowledge from simulation and analysis of models is a multi-step, iterative process requiring an array of software tools. NeuroML supports all stages of the model development life cycle (**Figure 1**), by providing a single model description format that interacts with a myriad of tools throughout the process. Researchers typically assemble ad-hoc sets of scripts, applications, and processes to help them in their investigations. In the absence of standardization, they must work with the specific model formats and APIs that each tool they use requires, and somehow convert model descriptions when using multiple applications in a toolchain. NeuroML addresses this issue by providing a common language for the use and exchange of models and their components between different simulation engines and modeling tools. The NeuroML ecosystem includes a large collection of software tools, both developed and maintained by the main NeuroML contributors (the 'core NeuroML tools and libraries:' jNeuroML, pyNeuroML, APIs) and those external applications that have added NeuroML support (**Figures 3 and 4a**, **Tables 3 and 4**).

The core NeuroML tools and libraries include APIs in several programming languages—Python, Java, C++, and MATLAB. These tools provide critical functionality to allow users to interact with NeuroML components and build models. Using these, researchers can build models from scratch, or read, modify, analyze, visualize, and simulate existing NeuroML models on supported simulation

**Table 2.** Index of standard NeuroMLv2 ComponentTypes (continued).

**ComponentTypes related to synapses**

| | | |
|---|---|---|
| alphaCurrentSynapse | alphaSynapse | blockingPlasticSynapse |
| doubleSynapse | expOneSynapse | expThreeSynapse |
| expTwoSynapse | gap Junction | gradedSynapse |
| linearGradedSynapse | silentSynapse | stdpSynapse |
| tsodyksMarkramDepFacMechanism | tsodyksMarkramDepMechanism | voltageConcDepBlockMechanism |

**ComponentTypes related to inputs**

| | | |
|---|---|---|
| compoundInput | compoundInputDL | poissonFiringSynapse |
| pulseGenerator | pulseGeneratorDL | rampGenerator |
| rampGeneratorDL | sineGenerator | sineGeneratorDL |
| spike | spikeArray | spike Generator |
| spikeGeneratorPoisson | spikeGeneratorRandom | spikeGeneratorRefPoisson |
| timedSynapticInput | transientPoissonFiringSynapse | voltage Clamp |
| voltageClampTriple | | |

**ComponentTypes related to networks**

| | | |
|---|---|---|
| connection | connectionWD | continuous Connection |
| continuousConnectionInstance | continuousConnectionInstanceW | continuous Projection |
| electrical Connection | electricalConnectionInstance | electricalConnectionInstanceW |
| electrical Projection | explicit Connection | explicitInput |
| input | inputList | inputW |
| instance | location | network |
| networkWithTemperature | population | population List |
| projection | rectangularExtent | region |
| synaptic Connection | synapticConnectionWD | |

**ComponentTypes related to model simulation**

| | | |
|---|---|---|
| Display | EventOutputFile | EventSelection |
| Line | OutputColumn | OutputFile |
| Simulation | | |

**ComponentTypes related to PyNN**

| | | |
|---|---|---|
| alphaCondSynapse | alphaCurrSynapse | EIF_cond_alpha_isfa_ista |
| EIF_cond_exp_isfa_ista | expCondSynapse | expCurrSynapse |
| HH_cond_exp | IF_cond_alpha | IF_cond_exp |
| IF_curr_alpha | IF_curr_exp | SpikeSourcePoisson |

platforms. Furthermore, developers can also use the core tools, libraries, and APIs to support NeuroML in their own applications.

The simulation platforms e.g. EDEN (*Panagiotou et al., 2022*), NEURON (*Hines and Carnevale, 1997*), along with other independently developed tools, form the next layer of the software ecosystem—providing extra functionality such as interactive model construction (e.g. neuroConstruct *Gleeson et al., 2007*), NetPyNE (*Dura-Bernal et al., 2019*), additional visualization (e.g. OSB *Gleeson et al., 2019b*), analysis (e.g. NeuroML-DB *Birgiolas et al., 2023*), data-driven validation (e.g. SciUnit *Gerkin et al., 2019*), and archival/sharing (e.g. OSB, NeuroML-DB). Indeed, OSB and NeuroML-DB are prime examples of how advanced neuroinformatics resources can be built on top of standards such as NeuroML.

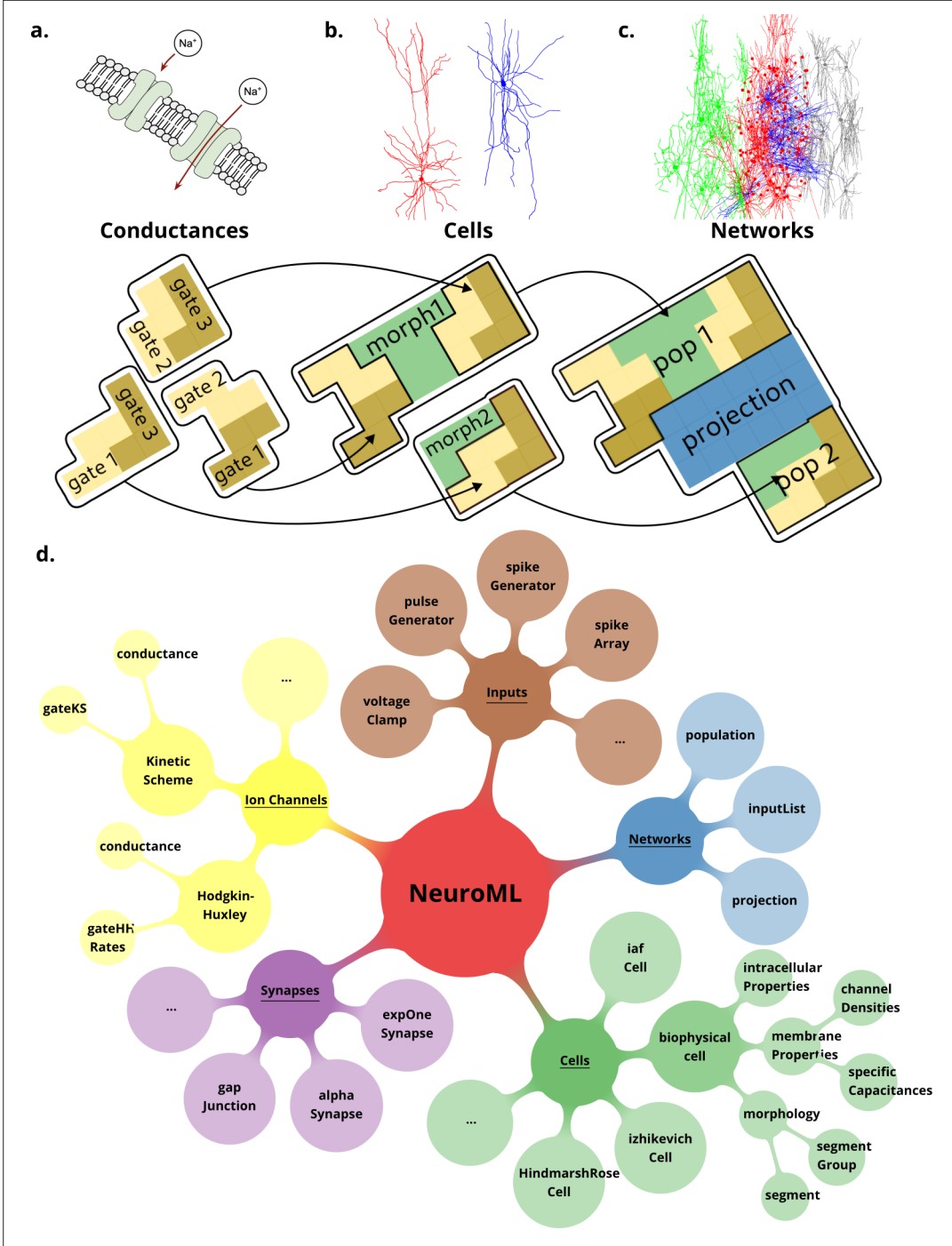

**Figure 2.** NeuroML is a modular, hierarchical format that supports multi-scale modeling. Elements in NeuroML are formally defined, independent, self-contained building blocks with hierarchical relationships between them. (**a**) Models of **ionic conductances** can be defined as a composition of gates, each with specific voltage (and potentially [Ca$^{2+}$]) dependence that controls the conductance. (**b**) Morphologically detailed **neuronal models** specify the 3D structure of the cells, along with passive electrical properties, and reference ion channels that confer membrane conductances. (**c**) Network models contain populations of these cells connected via synaptic projections. (**d**) A truncated illustration of the main categories of the NeuroMLv2 standard elements and their hierarchies. The standard includes commonly used model elements/building blocks that have been pre-defined for users: **Cells**: neuronal models ranging from simple spiking point neurons to biophysically detailed cells with multi-compartmental morphologies and active membrane conductances; **Synapses and ionic conductance models**: commonly used chemical and electrical synapse models (gap junctions), and multiple representations for

*Figure 2 continued on next page*

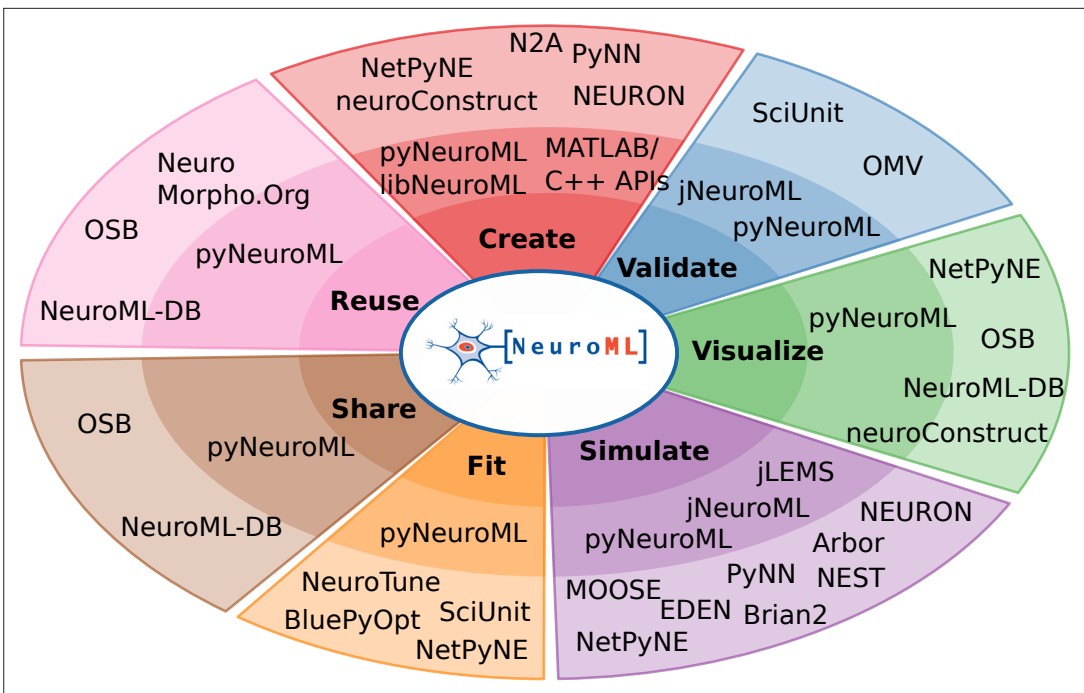

**Figure 3.** NeuroML compliant tools and their relation to the model life cycle. The inner circle shows the core NeuroML tools and libraries that are maintained by the NeuroML developers. These provide the functionality to read, modify, or create new NeuroML models, as well as to validate, analyze, visualize and simulate the models. The outermost layer shows NeuroML-compliant tools that have been developed independently to allow various interactions with NeuroML models. These complement the core tools by facilitating model creation, validation, visualization, simulation, fitting/optimization, sharing, and reuse. Further information on each of the tools shown here can be found in *Tables 3 and 4*.

*Table 5* lists interactive, step-by-step guides in the NeuroML documentation, which can be followed to learn the fundamental NeuroML concepts, as well as illustrate how NeuroML-compliant tools can be used to achieve specific tasks across the model development life cycle. In the following sections, we discuss the specific functionality available at each stage of model development.

## Creating NeuroML models

The structured declarative elements of NeuroMLv2, when combined with a procedural scripting language such as Python, provide a powerful and yet intuitive 'building block' approach to model construction. For this reason, Python is now the recommended language for interacting with NeuroML (*Figure 4*), although XML remains the primary serialization language for the format (i.e. for saving to disk and depositing in model repositories (*Figure 5*)). Python has emerged as a key programming language in science, including many areas of neuroscience (*Muller et al., 2015*). A Python-based NeuroML ecosystem ensures that users can take advantage of Python's features, and also use packages from the wider Python ecosystem in their work (e.g. Numpy (*Harris et al., 2020*), Matplotlib *Hunter, 2007*). pyNeuroML, the Python interface for working with NeuroML, is built on top of the Python NeuroML API, libNeuroML (*Vella et al., 2014*; *Sinha, 2023*; *Figure 4*).

As illustrated in *Figure 5*, Python can be used to combine different NeuroML components into a model. NeuroML supports several pathways for the creation of new models. Modelers may use

*Figure 2 continued*

ionic conductances; **Inputs**: to drive cell and network activity, e.g., current or voltage clamp, spiking background inputs; **Networks**: of populations (containing any of the aforementioned cell types), and projections. The full list of standard NeuroML elements can be found in *Tables 1 and 2*.

**a.**

**b.** Example Python usage

```
from neuroml import * # NeuroML API libNeuroML

newdoc = NeuroMLDocument(id="new_doc")
newcell = IafTauCell(id="cell_0",
    leak_reversal="-60mV", thresh="0mV",
    tau="5ms", reset="-70mV")
newdoc.add(newcell)

network = newdoc.add(Network, id="new_net",
    validate=False)
population = network.add(Population,
    id="new_pop", size=10,
    component=newcell.id)

# Helper method to ensure all parameters
# present and appropriate
newdoc.validate(recursive=True)
```

**c.** XML serialization

```
<neuroml id="new_doc">

  <iafTauCell id="cell_0"
      leakReversal="-60mV" thresh="0mV"
      reset="-70mV" tau="5ms"/>

  <network id="new_net">

      <population id="new_pop"
          component="cell_0" size="10"/>

  </network>

</neuroml>
```

**Figure 4.** The core NeuroML software stack, and an example NeuroML model created using the Python NeuroML tools. (**a**) The core NeuroML software stack consists of Java (blue) and Python (orange) based applications/libraries, and the LEMS model ComponentType definitions (green), wrapped up in a single package, pyNeuroML. Each of these modules can be used independently or the whole stack can be obtained by installing pyNeuroML with the default Python package manager, Pip: pip install pyneuroml. (**b**) An example of how to create a simple NeuroML model is shown, using the NeuroMLv2 Python API (libNeuroML) to describe a model consisting of a population of 10 integrate and fire point neurons (IafTauCell) in a network. The IafTauCell, Network, Population, and NeuroMLDocument model ComponentTypes are provided by the NeuroMLv2 standard. The underlying dynamics of the model are hidden from the user, being specified in the LEMS ComponentType definitions of the elements (see Methods). The simulator-independent NeuroML model description can be simulated on any of the supported simulation engines. (**c**) Extensible Markup Language (XML) serialization of the NeuroMLv2 model description shows the correspondence between the Python object model and the XML serialization.

**Table 3.** NeuroML software core tools and libraries, with a description of their scope, the main programming language they use (or other interaction means, e.g. Command Line Interface (CLI)), and links for more information.

| Tool | Language/interface | Description | URL |
|---|---|---|---|
| pyNeuroML | Python/CLI | Recommended Python library for NeuroML; provides pynml, primary command line tool for NeuroML | https://docs.neuroml.org/Userdocs/Software/pyNeuroML.html |
| libNeuroML | Python | Python API for NeuroML | https://docs.neuroml.org/Userdocs/Software/libNeuroML.html |
| NeuroMLlite | Python | High level library for creating NeuroML network models (beta) | https://docs.neuroml.org/Userdocs/Software/NeuroMLlite.html |
| PyLEMS | Python/CLI | Python API and simulator for LEMS | https://docs.neuroml.org/Userdocs/Software/pyLEMS.html |
| jLEMS | Java/CLI | Java API for LEMS and reference simulator | https://docs.neuroml.org/Userdocs/Software/jLEMS.html |
| org.neuroml.model | Java | Java API for NeuroML, DOI:10.5281/zenodo.5783290 | https://github.com/NeuroML/org.neuroml.model/ |
| org.neuroml.export | Java | Java API for translating NeuroML into different formats such as NEURON, DOI:10.5281/zenodo.1346272 | https://github.com/NeuroML/org.neuroml.export |
| org.neuroml.import | Java | Java API for importing formats into LEMS and NeuroML, DOI:10.5281/zenodo.5783295 | https://github.com/NeuroML/org.neuroml.import |
| jNeuroML | Java/CLI | Wraps jLEMS and all export/import packages and provides the jnml tool, DOI:10.5281/zenodo.593108 | https://docs.neuroml.org/Userdocs/Software/jNeuroML.html |
| NeuroML-C++ | C++ | C++ API for NeuroML | https://docs.neuroml.org/Userdocs/Software/NeuroML_API.html |
| NeuroML Toolbox | MATLAB | MATLAB NeuroML Toolbox | https://docs.neuroml.org/Userdocs/Software/MatLab.html |

elements included in the NeuroML standard, re-use user-defined NeuroML model elements from other models, or define completely new model elements using LEMS (*Figure 5*) (see section on extending NeuroML below). It is common for models to use a combination of these strategies, e.g., *Gurnani and Silver, 2021*; *Kriener et al., 2022*; *Cayco-Gajic et al., 2017*, highlighting the flexibility provided by the modular design of NeuroML. NeuroML APIs support all of these workflows. The Python tools also include many additional higher-level utilities to speed up model construction, such as factory functions, type hints, and convenience functions for building complex multi-compartmental neuron models (*Figure 6*).

For the construction of complex 3D circuit models, or for users who are not experienced with Python, a range of NeuroML-compliant online and standalone applications with graphical user interfaces are available. These include NetPyNE's interactive web interface (*Dura-Bernal et al., 2019*) (which is available on the latest version of OSB (https://v2.opensourcebrain.org)) and neuroConstruct (*Gleeson et al., 2007*) which can export models directly into NeuroML and LEMS. These applications can be used to build and simulate new NeuroML models without requiring programming. Thus, users can take advantage of the individual features provided by these applications to generate NeuroML-compliant models and model elements.

## Validating NeuroML models

Ensuring a model is 'valid' can have different meanings at different stages of the life cycle—from checking whether the source files are in the correct format, to ensuring the model reproduces a significant feature of its biological counterpart. NeuroML's hierarchical, well-defined structure allows users to check their model descriptions for correctness at multiple levels (*Figure 7*), in a manner similar to

**Table 4.** Tools in the wi main programming language they use (or other interaction means, e.g. through a web browser, Graphical User Interface (GUI) or Command Line Interface (CLI)), and links for more information.

| Tool | Language/interface | Description | URL |
|---|---|---|---|
| **Simulation engines** | | | |
| NEURON | Python/Hoc/CLI/GUI | Empirically-based simulations of neurons and networks of neurons | https://docs.neuroml.org/Userdocs/Software/Tools/NEURON.html |
| NetPyNE | Python/web | Package to facilitate the development, parallel simulation, analysis, and optimization of biological neuronal networks using the NEURON simulator. Also has a graphical web interface, NetPyNE-UI | https://docs.neuroml.org/Userdocs/Software/Tools/NetPyNE.html |
| EDEN | NeuroML | NeuroML-based neural simulator | https://docs.neuroml.org/Userdocs/Software/Tools/EDEN.html |
| MOOSE | Python | The Multiscale Object-Oriented Simulation Environment is the base and numerical core for large, detailed multi-scale simulations that span computational neuroscience and systems biology. Based on a reimplementation of the GENESIS 2 core. | https://docs.neuroml.org/Userdocs/Software/Tools/MOOSE.html |
| PyNN | Python | A simulator-independent language for building neuronal network models | https://docs.neuroml.org/Userdocs/Software/Tools/PyNN.html |
| NEST | Python/SLI | Simulator for spiking neural network models focusing on dynamics, size, and structure of neural systems | https://docs.neuroml.org/Userdocs/Software/Tools/NEST.html |
| Brian2 | Python | Easy to learn and use simulator for spiking neural networks | https://docs.neuroml.org/Userdocs/Software/Tools/Brian.html |
| Arbor | Python | A multi-compartment neuron simulation library | https://docs.neuroml.org/Userdocs/Software/Tools/Arbor.html |
| N2A | Java/GUI | Language and IDE for writing and simulating models | https://docs.neuroml.org/Userdocs/Software/Tools/N2A.html |
| **Databases** | | | |
| OSB | Web | Resource for sharing and collaboratively developing computational models of neural systems | https://www.opensourcebrain.org/ |
| NeuroML-DB | Web | NeuroML database of cell and channel models | https://neuroml-db.org/ |
| **Other tools** | | | |
| OMV | Python | Open Source Brain Model Validation framework | https://github.com/OpenSourceBrain/osb-model-validation |
| SciUnit | Python | Data driven unit testing framework | https://github.com/scidash/sciunit |
| BluePyOpt | Python | Blue Brain Python Optimization Library | https://bluepyopt.readthedocs.io/ |
| NeuroTune | Python | Package for fitting/optimization of NeuroML models | https://github.com/NeuralEnsemble/neurotune |
| PyElectro | Python | Electrophysiology analysis package | https://github.com/NeuralEnsemble/pyelectro |

**Table 5.** Step-by-step guides for using NeuroML illustrating the various stages of the model development life cycle.

These include Introductory guides aimed at teaching the fundamental NeuroML concepts, Advanced guides illustrating specific modeling workflows, and Walkthrough guides discussing the steps required for converting models to NeuroML. An updated list is available at http://neuroml.org/gettingstarted.

| Link | Description | Model life cycle stages |
|---|---|---|
| Introductory guides | | |
| Guide 1 | Create and simulate a simple regular spiking Izhikevich neuron in NeuroML | Create, Validate, Simulate |
| Guide 2 | Create a network of two synaptically connected populations of Izhikevich neurons | Create, Validate, Visualize, Simulate |
| Guide 3 | Build and simulate a single compartment Hodgkin-Huxley neuron | Create, Validate, Visualize, Simulate |
| Guide 4 | Create and simulate a multi compartment hippocampal OLM neuron | Create, Validate, Visualize, Simulate |
| Advanced guides | | |
| Guide 5 | Create novel NeuroML models from components on NeuroML-DB | Reuse, Create, Validate, Simulate |
| Guide 6 | Optimize/fit NeuroML models to experimental data | Create, Validate, Simulate, Fit |
| Guide 7 | Extend NeuroML by creating a novel model type in LEMS | Create, Simulate |
| Walkthroughs | | |
| Guide 8 | Guide to converting cell models to NeuroML and sharing them on Open Source Brain | Create, Validate, Simulate, Share |
| Guide 9 | Conversion of *Ray et al., 2020* | Create, Validate, Visualize, Simulate, Share |

multi-level testing in software development. Importantly, most of the validation tests in NeuroML are run on the models' NeuroML descriptions *prior to simulation*.

A first level of validation checks the structure of individual model elements against their formal specifications contained in the NeuroML standard. The standard includes information on the parameters of each model element, restrictions on parameter values, their allowed units, their cardinality, and the location of the model element in the model hierarchy—i.e., parent/children relationships. A second level of validation includes a suite of semantic and logical checks. For example, at this level, a model of a multi-compartmental cell can be checked to ensure that all segments referenced in segment groups (e.g. the group of dendritic segments) have been defined, and only defined once with unique identifiers. A list of validation tests currently included in the NeuroML core tools can be found in *Table 6*. These can be run against NeuroML files at the command line or programmatically in Python (*Figure 6*).

A key advantage of using the NeuroML2/LEMS framework is that dimensions and units are inbuilt into LEMS descriptions. This enables automated conversions of units, unit checking, together with the validation of equations. Any expressions in models which are dimensionally inconsistent will be highlighted at this stage. Note that LEMS handles unit conversions internally—modelers have flexibility in how they enter the *units* of parameter values (e.g. specifying conductance density in $S/m^2$ or $mS/cm^2$) in the NeuroML files, with the underlying LEMS definitions ensuring that a consistent set of *dimensions* are used in model equations (*Cannon et al., 2014*). LEMS then takes care of mapping the entered units to the target simulator's preferred units. This makes model definition, inspection, use, extension, and translation easier and less error-prone.

Once the set of NeuroML files are validated, the model can be simulated, and checks can be made to test whether execution produces consistent results (e.g. firing rate of neurons in a given population) across multiple simulators (or versions of the same simulator). For this, the OSB Model Validation (OMV) framework has been developed (*Gleeson et al., 2019b*). This framework can automatically check that the output (e.g. spike times) of a NeuroML model running on a given simulator is within an allowed tolerance of the expected value. OMV has been applied to NeuroML models that have been shared on OSB, to test consistent behavior of models as the models themselves, and all supported

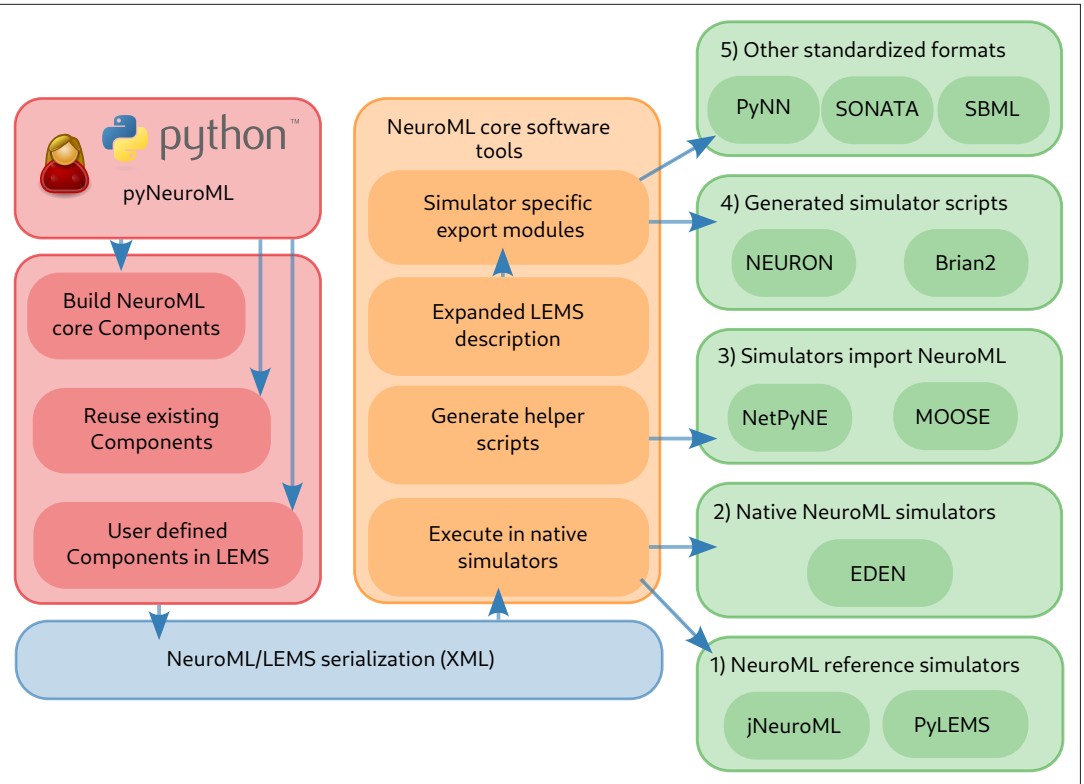

**Figure 5.** Workflow showing how to create and simulate NeuroML models using Python. The Python API can be used to create models which may include elements built from scratch from the NeuroML standard, re-use elements from previously created models, or create new components based on novel model definitions expressed in LEMS (red). The generated model elements are saved in the default XML-based serialization (blue). The NeuroML core tools and libraries (orange) include modules to import model descriptions expressed in the XML serialization, and support multiple options for how simulators can execute these models (green). These include: (1) execution of the NeuroML models by reference simulators; (2) execution by other independently developed simulators that natively support NeuroML, such as EDEN; (3) generation of Python 'import scripts' which allow NeuroML models to be imported (and converted to internal formats) by simulators which support this; (4) fully expanding the LEMS description of the models, which can be mapped to generated simulator specific scripts for target simulators; (5) mapping to other standardized formats in neuroscience and systems biology.

simulators, are updated. This has proven to be a valuable process for ensuring uniform usage and interpretation of NeuroML across the ecosystem of supporting tools.

A final level of validation concerns checking whether the model elements have emergent features that are in line with experimentally observed behavior of the biological equivalents. NeuronUnit (*Gerkin et al., 2019*), a SciUnit (*Omar et al., 2014*) package for data-driven unit testing and validation of neuronal and ion channel models, is also fully NeuroML compliant, and also supports automated validation of NeuroML models shared on NeuroML-DB and OSB.

## Visualizing/analyzing NeuroML models

Multiple visualization, inspection, and analysis tools are available in the NeuroML software ecosystem. Since NeuroML models have a fixed, well-defined structure, NeuroML libraries can extract all information from their descriptions. This information can be used by modelers and their programs/tools to run automated programmatic analyses on models.

pyNeuroML includes a range of ready-made inspection utilities for users (*Figure 6*) that can be used via Python scripts, interactive Jupyter Notebooks, and command line tools. Examining the structure of cell and network models with 2D and 3D views is important for manual validation and to compare them to their biological counterparts. Graphical views of cell model morphology and the 3-dimensional network layout (*Figure 8*), population and connectivity matrices/graphs at different levels (*Figure 9*), and model summaries can all be generated (*Figure 10*). In addition to these inspection functions, a

**Create (using Python API)**

```python
from neuroml import *

# Create a container document
doc = NeuroMLDocument(id="network0")

# Add single exponential synapse model
syn0 = doc.add("ExpOneSynapse", id="syn0", gbase="65nS", erev="0mV", tau_decay="3ms")

# Reuse existing ion channel model
doc.add("IncludeType", href="Na_chan.channel.nml")

# Create a cell with 3D morphology using the Cell ComponentType
cell = doc.add("Cell", id="olm", neuro_lex_id="NLXCELL:091206") # Hippocampal CA1 OLM cell
cell.set_init_memb_potential("-67mV")
cell.set_resistivity("0.15 kohm_cm")
cell.add_channel_density(doc, cd_id="na_all", cond_density="10 mS_per_cm2",
                         ion_channel="Na_chan", ion_chan_def_file="Na.channel.nml",
                         erev="50mV", ion="na")
cell.add_unbranched_segment_group("soma_group")
soma_0 = cell.add_segment(prox=[0, 0, 0, 10], dist=[0, 10, 0, 10], name="Seg0_soma_0",
                          group_id="soma_group", seg_type="soma")
```

| API examples | Command line usage examples |
|---|---|

**Validate**

```python
validate_neuroml2("file.nml")
doc.validate(recursive=True)
```
```
> pynml "file.nml" -validate
```

**Inspect and visualize**

```python
element.info()
summary(doc)
nml2_to_png(doc)
nml2_to_svg(doc)
generate_nmlgraph(doc)

plot_2D(cell)
plot_interactive_3d(cell)
plot_interactive_3d(network)

plot_channel_densities(cell)
```
```
> pynml-summary "file.nml"
> pynml -png "file.nml"
> pynml -svg "file.nml"
> pynml "file.nml" -graph
> pynml "file.nml" -matrix 1

> pynml-plotmorph "cell.nml"
> pynml-plotmorph -interactive3d "cell.nml"
> pynml-plotmorph -interactive3d "net.nml"
> pynml-channelanalysis "channel.nml"
> pynml-plotchan "cell.nml"
```

**Simulate**

```python
run_lems_with_jneuroml("sim.xml")
run_lems_with_jneuroml_neuron("sim.xml")
run_lems_with_jneuroml_netpyne("sim.xml")
run_on_nsg("jneuroml_neuron", "sim.xml")
...
```
```
> pynml "sim.xml"
> pynml "sim.xml" -neuron -run
> pynml "sim.xml" -netpyne -run
```

**Share and archive**

```python
create_combine_archive("sim.xml")
```
```
> pynml-archive "neuron.cell.nml"
```

**Figure 6.** PyNeuroML provides Python functions and command line utilities supporting all stages of the model life cycle.

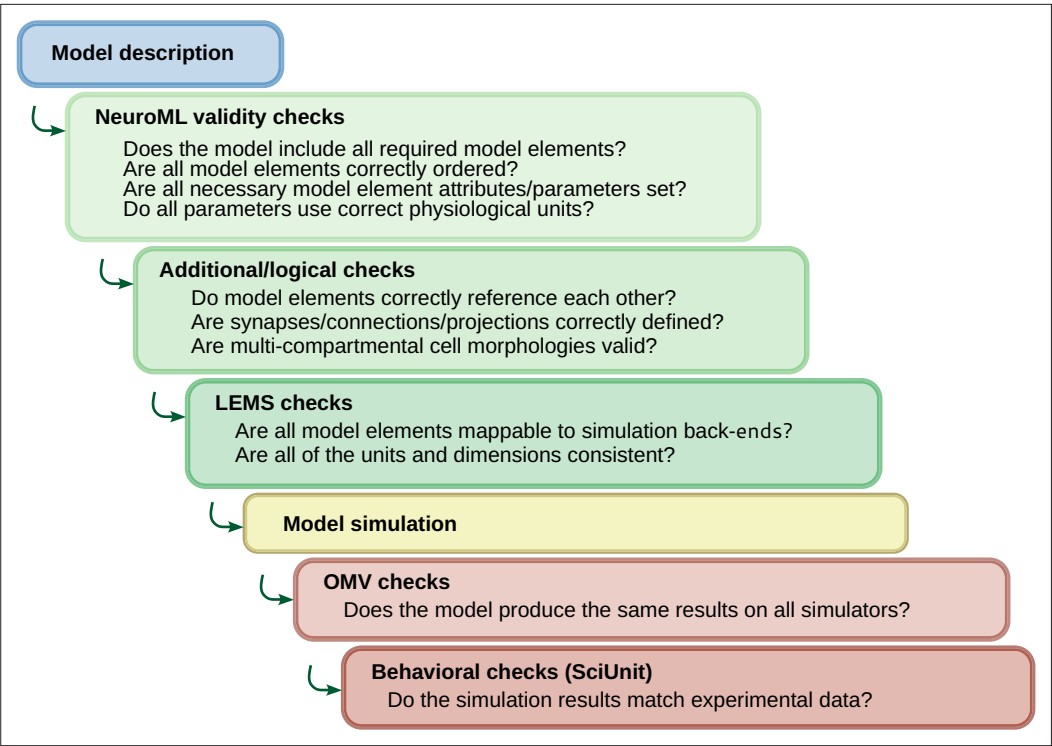

**Figure 7.** NeuroML model development incorporates multi-level validation of models. Checks are performed on the model descriptions (blue) before simulation using validation at both the NeuroML and LEMS levels (green). After the models are simulated (yellow), further checks can be run to ensure the output is in line with expected behavior (brown). The OSB Model Validation (OMV) framework can be used to ensure consistent behavior across simulators, and comparisons can be made of model activity to their biological equivalents using SciUnit.

variety of utilities for the inspection of NeuroML descriptions of electrophysiological properties of membrane conductances and their spatial distribution over the neuronal membrane are also provided (*Figure 10*).

The graphical applications included in the NeuroML ecosystem (e.g. neuroConstruct, NeuroML-DB, OSB (v1 [https://v1.opensourcebrain.org] and v2), NetPyNE, and Arbor-GUI) also provide many of their own analysis and visualization functions. OSBv1, for example, supports automated 3D visualization of networks and cell morphologies, network connectivity graphs and metrics, and advanced model inspection features (*Gleeson et al., 2019b*; *Figure 8b*). On OSBv2, NetPyNE provides advanced graphical plotting and analysis facilities (*Figure 8c*). A complete JupyterLab (https://jupyter.org/) interface is also included in OSBv2 for Python scripting, allowing interactive notebooks to be created and shared, mixing scripting and graphical elements, including those generated by pyNeuroML. NeuroML-DB also provides information on electrophysiology, morphology, and the simulation aspects of neuronal models (*Birgiolas et al., 2023*; *Figure 10a*). In general, any NeuroML-compliant application can be used to inspect and analyze elements of NeuroML models, each having their own distinct advantages.

## Simulating NeuroML models

Users can simulate NeuroML models using a number of simulation engines without making any changes to their models. This is because the NeuroML/LEMS descriptions of the models are simulator independent and can be translated to simulator specific formats. pyNeuroML facilitates access to all available simulation options, both from the command line and using function calls in Python scripts when using the Python API (*Figure 6*).

Simulation engines can be classified into five broad categories (*Figure 5*):

1. reference NeuroML/LEMS simulators.
2. independently developed simulators that natively support NeuroML.

**Table 6.** Listing of validation tests run by NeuroML.

| Test | Description |
|---|---|
| **Schema tests** | |
| Check names | Check that names of all elements, attributes, parameters match those provided in the schema |
| Check types | Check that the types of all included elements |
| Check values | Check that values follow given restrictions |
| Check inclusion | Check that required elements are included |
| Check cardinality | Check the number of elements |
| Check hierarchy | Check that child/children elements are included in the correct parent elements |
| Check sequence order | Check that child/children elements are included in the correct order |
| **Additional tests** | |
| Check top level ids | Check that top level (root) elements have unique ids |
| Check Network level ids | Check that child/children of the Network element have unique ids |
| Check Cell Segment ids | Check that all Segments in a Cell have unique ids |
| Check single Segment without parent | Check that only one Segment is without parents (the soma Segment) |
| Check SegmentGroup ids | Check that all SegmentGroups in a Cell have unique ids |
| Check Member segment ids exist | Check that Segments referred to in SegmentGroup Members exist |
| Check SegmentGroup definition | Check that SegmentGroups being referenced are defined |
| Check SegmentGroup definition order | Check that SegmentGroups are defined before being referenced |
| Check included SegmentGroups | Check that SegmentGroups referenced by Include elements of other SegmentGroups exist |
| Check numberInternalDivisions | Check that SegmentGroups define numberInternalDivisions (used by simulators to discretize un-branched branches into compartments for simulation) |
| Check included model files | Check that model files included by other files exist |
| Check Population component | Check that a component id provided to a Population exists |
| Check ion channel exists | Check that an ion channel used to define a ChannelDensity element exists |
| Check concentration model species | Check that the species used in ConcentrationModel elements are defined |
| Check Population size | Check that the size attribute of a PopulationList matches the number of defined Instances |
| Check Projection component | Check that Populations used in the Projection elements exist |
| Check Connection Segment | Check that the Segment used in Connection elements exist |
| Check Connection pre/post cells | Check that the pre- and post-synaptic cells used in Connection elements exist and are correctly specified |
| Check Synapse | Check that the Synapse component used in a Projection element exists |
| Check root id | Check that the root Segment in a Cell morphology has id 0 |

3. simulators that import/translate NeuroML to their own internal formats.
4. simulators that are supported through generation of simulator-specific scripts by the core NeuroML tools.
5. export to other standardized formats which may allow simulation/analysis in other packages.

Each simulation engine supports a different set of features that NeuroML users can take advantage of (*Table 7*). For example, the reference NeuroML and LEMS simulators, jNeuroML, jLEMS, and PyLEMS, can simulate all LEMS models and most NeuroML models. They cannot, however, simulate

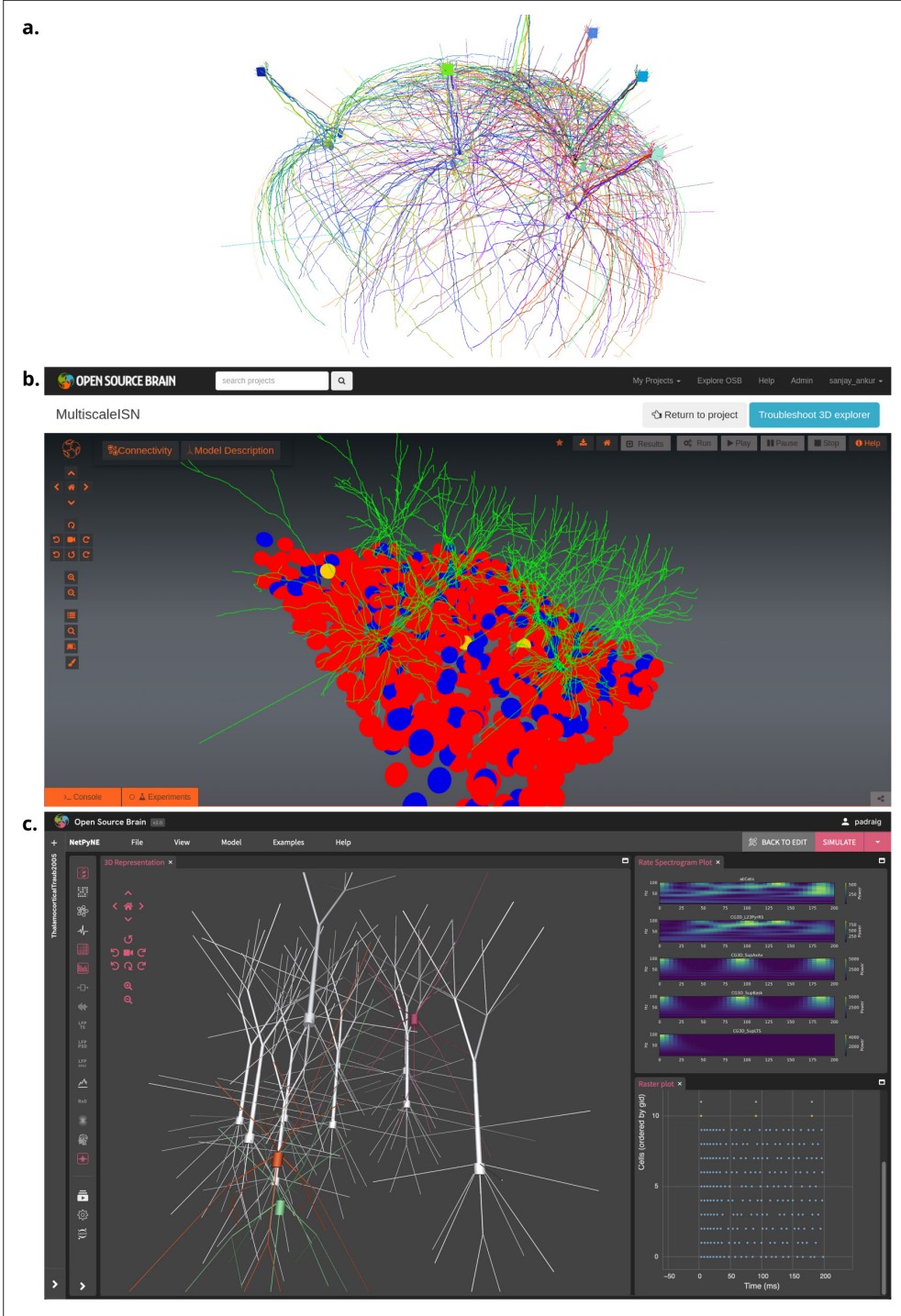

**Figure 8.** Visualization of detailed neuronal morphology of neurons and networks together with their functional properties (results from model simulation) enabled by NeuroML. (**a**) Interactive 3-D (VisPy (**Campagnola, 2023**) based) visualization of an olfactory bulb network with detailed mitral and granule cells (**Migliore et al., 2014**), generated using pyNeuroML. (**b**) Visualization of an inhibition stabilized network based on **Sadeh et al., 2017** using Open Source Brain (OSB) version 1 (**Gleeson et al., 2019b**). (**c**) Visualization of 3D network of simplified multi-compartmental cortical neurons (from **Traub et al., 2005**, imported as NeuroML **Gleeson, 2019a**) and simulated spiking activity using NetPyNE's GUI (**Dura-Bernal et al., 2019**), which is embedded in OSB version 2.

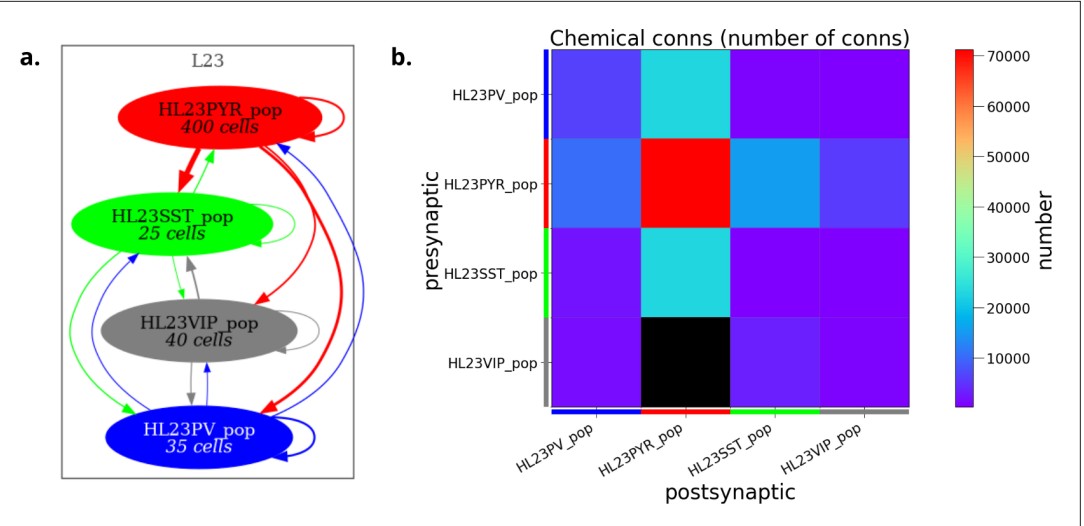

**Figure 9.** Analysis and visualization of network connectivity from NeuroML model descriptions *prior to simulation*. Network connectivity schematic (**a**) and connectivity matrix (**b**) for a half scale implementation of the human layer 2/3 cortical network model (*Yao et al., 2022*) generated using pyNeuroML.

multi-compartmental models, and users should opt for a simulator that does, e.g., NEURON (*Hines and Carnevale, 1997*) or EDEN (*Panagiotou et al., 2022*).

Another criteria that is relevant when choosing a simulation engine is the efficiency of simulation. Simulation engines implement different computing techniques—e.g., NetPyNE, Arbor, and EDEN support parallel execution on clusters and super computers via MPI—to enable simulation of large-scale models. Thus, for efficient large-scale simulation, users may prefer one of these simulation engines.

The preferred programming language for working with NeuroML is Python (*Muller et al., 2015*). A Python-based ecosystem ensures that automated simulation of models can easily be carried out either using scripts, or the command line tools. Utilities to enable the execution of simulations on dedicated supercomputing resources, such as the Neuroscience Gateway (NSG) (*Sivagnanam, 2013*;

**Table 7.** Features supported by NeuroML in different simulation engines.
Note: the simulators themselves may support more features, but these have not been mapped onto by the NeuroML tools. Abstract cell models: abstract cell models included in the NeuroML standard (see *Table 1*). Single compartmental cells: neuronal models that include a single compartment (these engines do not support multi-compartmental cells). Multiple compartmental cells: neuronal models that include multiple compartments. Conductance-based models: models that support ionic conductances. Parallel execution: engines that support parallel execution using MPI/GPUs. Y: full support; N: no support; L: limited support in NeuroML toolchain.

| Tool | Abstract cell models | Single compartment cells | Multiple compartment cells | Conductance-based models | Parallel execution |
|------|---------------------|--------------------------|----------------------------|--------------------------|--------------------|
| jNeuroML/pyNeuroML | Y | Y | N | Y | N |
| NEURON | Y | Y | Y | Y | N |
| NetPyNE | Y | Y | Y | Y | Y |
| EDEN | Y | Y | Y | Y | Y |
| MOOSE | Y | Y | L | Y | N |
| PyNN | Y | Y | L | L | Y |
| NEST | Y | Y | N | N | Y |
| Brian2 | Y | Y | Y | Y | L |
| Arbor | L | Y | Y | L | Y |

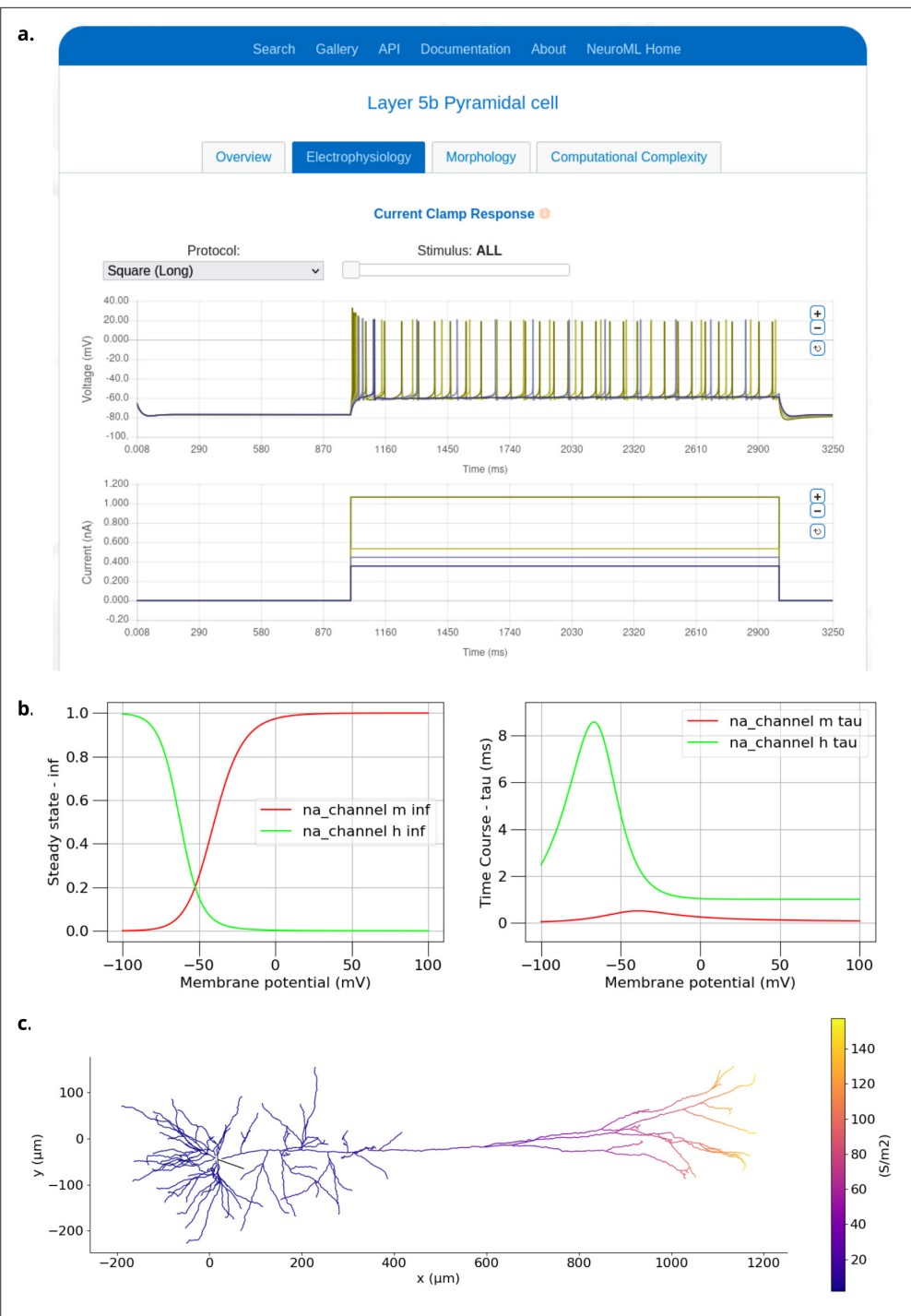

**Figure 10.** Examples of visualizing biophysical properties of a NeuroML model neuron. (**a**) Electrophysiological properties generated by the NeuroML-DB web-based platform (**Birgiolas et al., 2023**). (Plots show four superimposed voltage traces in the top panel and corresponding current injection traces below). (**b**) Example plots of steady states of activation (na_channel na_m inf) and inactivation (na_channel na_h inf) variables and their time courses (na_channel na_m tau and na_channel na_h tau) for the Na channel from the classic Hodgkin Huxley model (**Hodgkin and Huxley, 1952**). (**c**) The distribution of the peak conductances for the Ih channel over a layer 5 Pyramidal cell (**Hay et al., 2011**). Both (**b**) and (**c**) were generated using the analysis features in pyNeuroML, and similar functionality is also available in OSBv1 (**Gleeson et al., 2019b**).

http://www.nsgportal.org/) are also available within the ecosystem. OSBv1 takes advantage of these to support the submission of NeuroML model simulation jobs using the NEURON simulator on NSG. NetPyNE also includes parallel execution of simulations, batch processing, and parameter exploration features, and its deployment on OSBv2 allows users to easily access these features on a scalable, cloud-based platform. Finally, the JupyterLab environment on OSBv2 contains all of the core NeuroML tools and various simulation engines as pre-installed software packages, ready to use.

## Optimizing NeuroML models

Development of biologically detailed models of brain function requires that components and emergent properties match the behavior of the corresponding biology as closely as possible. Thus, fitting neurons and networks to experimental data is a critical step in the model life cycle (*Rossant et al., 2011*; *Druckmann et al., 2007*). pyNeuroML promotes data-driven modeling by providing functions to fit and optimize NeuroML models against experimental data. It includes the NeuroMLTuner module (https://pyneuroml.readthedocs.io/en/development/pyneuroml.tune.html), which builds on the Neurotune package (https://github.com/NeuralEnsemble/neurotune; *Vella and Gleeson, 2023*) for tuning and optimizing NeuroML models against data using evolutionary computation techniques. This module allows users to select a set of weighted features from their data to calculate the fitness of populations of candidate models. In each generation, the fittest models are found and mutated to create the next generation of models, until a set of models that best exhibit the selected data features are isolated (see Guide 6 in *Table 5*) (https://docs.neuroml.org/Userdocs/OptimisingNeuroMLModels.html).

The NeuroML ecosystem includes multiple tools that also provide model fitting features. The Blue Brain Python Optimisation Library (BluePyOpt) (*Van Geit et al., 2016*), an extensible framework for data-driven model parameter optimization, supports exporting optimized models to NeuroML files (https://github.com/BlueBrain/BluePyOpt/blob/master/examples/neuroml/neuroml.ipynb). Similar to pyNeuroML, NetPyNE also uses the inspyred Python package (https://github.com/aarongarrett/inspyred; *Sinha and Garrett, 2024*) to provide evolutionary computation-based model optimization features (*Dura-Bernal et al., 2019*).

## Sharing NeuroML models

The NeuroML ecosystem includes the advanced web-based model sharing platforms NeuroML-DB (*Birgiolas et al., 2023*; https://neuroml-db.org) and OSB (*Gleeson et al., 2019b*). These resources have been designed specifically for the dissemination of models and model elements standardized in NeuroML. The OSB platform also supports visualization, analysis, simulation, and development of NeuroML models. Researchers can create shared, collaborative NeuroML projects on it and can take advantage of the in-built automated visualization and analysis pipelines to explore and re-use models and their components. Whereas version 1 (OSBv1) focused on providing an interactive 3D interface for running pre-existing NeuroML models (e.g. sourced from linked GitHub repositories) (*Gleeson et al., 2019b*), OSBv2 provides cloud-based workspaces for researchers to construct NeuroML-based computational models as well as analyze, and compare them to, the experimental data on which they are based, thus facilitating data-driven computational modeling. *Table 8* provides a list of stable, well-tested NeuroML compliant models from brain regions including the neocortex, cerebellum, and hippocampus, which have been shared on OSB.

NeuroML-DB aims to promote the uptake of standardized NeuroML models by providing a convenient location for archiving and exploration. It includes advanced database search functions, including ontology-based search (*Birgiolas et al., 2015*), coupled with pre-computed analyses on models' electrophysiological and morphological properties, as well as an indication of the relative speed of execution of different models.

NeuroML's modular nature ensures that models and their components can be easily shared with others through standard code sharing resources. The simplest way of sharing NeuroML models and components is to make their Python descriptions or their XML serializations available through these resources. Indeed, it is straightforward to make Python descriptions or the XML serializations available via different file, code (GitHub, GitLab), model sharing (ModelDB *Migliore et al., 2003*; *McDougal et al., 2017*), and archival (Zenodo, Open Science Framework) platforms, just like any other code/data produced in scientific investigations. Complex models with many components, spanning multiple files,

**Table 8.** Listing of NeuroML models and example repositories.

| Model | Description | URL |
| --- | --- | --- |
| **Neocortex** | | |
| *Billeh et al., 2020* | Morphologically detailed and point neuron models based on electrophysiological recordings from visual cortex neurons | https://github.com/OpenSourceBrain/AllenInstituteNeuroML |
| *Brunel, 2000* | Spiking network illustrating balance between excitation and inhibition | https://github.com/OpenSourceBrain/Brunel2000 |
| *Hay et al., 2011* | Layer 5 pyramidal cell model constrained by somatic and dendritic recordings | https://github.com/OpenSourceBrain/L5bPyrCellHayEtAl2011 |
| *Izhikevich, 2004* | Spiking neuron model reproducing wide range of neuronal activity | https://github.com/OpenSourceBrain/IzhikevichModel |
| *Markram et al., 2015* | Cell models from Neocortical Microcircuit of Blue Brain Project | https://github.com/OpenSourceBrain/BlueBrainProjectShowcase |
| *Pospischil et al., 2008* | HH-based models for different classes of cortical and thalamic neurons | https://github.com/OpenSourceBrain/PospischilEtAl2008 |
| *Potjans and Diesmann, 2014* | Microcircuit model of sensory cortex with 8 populations across 4 layers | https://github.com/OpenSourceBrain/PotjansDiesmann2014 |
| *Dura-Bernal et al., 2017* | Model of mouse primary motor cortex (M1) | https://github.com/OpenSourceBrain/M1NetworkModel |
| *Sadeh et al., 2017* | Point neuron model of Inhibition Stabilized Network | https://github.com/OpenSourceBrain/SadehEtAl2017-InhibitionStabilizedNetworks |
| *Smith et al., 2013* | Layer 2/3 cell model used to investigate dendritic spikes | https://github.com/OpenSourceBrain/SmithEtAl2013-L23DendriticSpikes |
| *Traub et al., 2005* | Single column network model containing 14 cell populations from cortex and thalamus | https://github.com/OpenSourceBrain/Thalamocortical |
| *Bahl et al., 2012* | A set of reduced models of layer 5 pyramidal neurons | https://github.com/OpenSourceBrain/BahlEtAl2012_ReducedL5PyrCell |
| *Wilson and Cowan, 1972* | A classic rate-based model describing the dynamics and interactions between the excitatory and inhibitory populations of neurons | https://github.com/OpenSourceBrain/WilsonCowan |
| *Garcia Del Molino et al., 2017* | Rate-based model showing paradoxical response reversal of top-down modulation in cortical circuits with three interneuron types | https://github.com/OpenSourceBrain/del-Molino2017 |
| *Mejias et al., 2016* | A rate-based model simulating the dynamics of a cortical laminar structure across multiple scales: intralaminar, interlaminar, interareal and whole cortex | https://github.com/OpenSourceBrain/MejiasEtAl2016 |
| **Cerebellum** | | |
| *Maex and Schutter, 1998* | Cerebellar granule cell | https://github.com/OpenSourceBrain/GranuleCell |
| *Cayco-Gajic et al., 2017* | Cerebellar granule cell layer network | https://github.com/SilverLabUCL/MF-GC-network-backprop-public |
| *Maex and Schutter, 1998* | 3D Cerebellar granule cell layer network | https://github.com/OpenSourceBrain/GranCellLayer |
| *Solinas et al., 2007* | Cerebellar Golgi cell model | https://github.com/OpenSourceBrain/SolinasEtAl-GolgiCell |
| *Vervaeke et al., 2010* | Electrically connected cerebellar Golgi cell network model | https://github.com/OpenSourceBrain/VervaekeEtAl-GolgiCellNetwork |
| **Hippocampus** | | |
| *Bezaire et al., 2016* | Full scale network model of CA1 region of hippocampus | https://github.com/mbezaire/ca1 |

*Table 8 continued on next page*

*Table 8 continued*

| Model | Description | URL |
|---|---|---|
| *Ferguson et al., 2013* | Parvalbumin-positive interneuron from CA1, based on Izhikevich cell model | https://github.com/OpenSourceBrain/FergusonEtAl2013-PVFastFiringCell |
| *Ferguson et al., 2014* | Pyramidal cell from CA1, based on Izhikevich cell model | https://github.com/OpenSourceBrain/FergusonEtAl2014-CA1PyrCell |
| *Migliore et al., 2005* | Multi-compartmental model of pyramidal cell from CA1 region of hippocampus | https://github.com/OpenSourceBrain/CA1PyramidalCell |
| *Pinsky and Rinzel, 1994* | Simplified model of CA3 pyramidal cell | https://github.com/OpenSourceBrain/PinskyRinzelModel |
| *Wang and Buzsáki, 1996* | Hippocampal interneuronal network model exhibiting gamma oscillations | https://github.com/OpenSourceBrain/WangBuzsaki1996 |
| Olfactory bulb | | |
| *Migliore et al., 2014* | Large-scale 3D olfactory bulb network with detailed mitral cells and granule cells | https://github.com/OpenSourceBrain/MiglioreEtAl14_OlfactoryBulb3D |
| Invertebrate | | |
| *Hodgkin and Huxley, 1952* | Classic investigation of the ionic basis of the action potential | https://github.com/openworm/hodgkin_huxley_tutorial |
| *FitzHugh, 1961* | Simplified form of Hodgkin Huxley model | https://github.com/OpenSourceBrain/FitzHugh-Nagumo |
| *Prinz et al., 2004* | Pyloric network of the lobster stomatogastric ganglion system | https://github.com/OpenSourceBrain/PyloricNetwork |
| *Boyle and Cohen, 2008* | Model of body wall muscle from *C. elegans* | https://github.com/openworm/muscle_model |
| *Gleeson et al., 2018* | A multiscale framework for modeling the nervous system of *C. elegans* | https://github.com/openworm/c302 |
| General | | |
| *Morris and Lecar, 1981* | Two dimensional reduced neuron model with calcium and potassium conductances | https://github.com/OpenSourceBrain/MorrisLecarModel |
| *Hindmarsh and Rose, 1984* | A simplified point cell model which captures complex firing patterns of single neurons, such as periodic and chaotic bursting | https://github.com/OpenSourceBrain/HindmarshRose1984 |
| Showcases | | |
| NEST Showcase | Examples of interactions with simulator NEST | https://github.com/OpenSourceBrain/NESTShowcase |
| PyNN Showcase | Examples of interactions between NeuroML and PyNN | https://github.com/OpenSourceBrain/PyNNShowcase |
| NetPyNE Showcase | Examples of interactions between NeuroML and NetPyNE | https://github.com/OpenSourceBrain/NetPyNEShowcase |
| SBML Showcase | Examples of interactions between NeuroML and SBML | https://github.com/OpenSourceBrain/SBMLShowcase |
| Brian Showcase | Examples of interactions between NeuroML and Brian | https://github.com/OpenSourceBrain/BrianShowcase |
| MOOSE Showcase | Examples of interactions between NeuroML and MOOSE | https://github.com/OpenSourceBrain/MOOSEShowcase |
| Arbor Showcase | Examples of interactions between NeuroML and Arbor | https://github.com/OpenSourceBrain/ArborShowcase |
| EDEN Showcase | Examples of interactions between NeuroML and EDEN | https://github.com/OpenSourceBrain/EDENShowcase |
| The Virtual Brain Showcase | Examples of interactions between NeuroML and TVB | https://github.com/OpenSourceBrain/TheVirtualBrainShowcase |

*Table 8 continued*

| Model | Description | URL |
|-------|-------------|-----|
| NEURON Showcase | Examples of interactions between NeuroML and NEURON | https://github.com/OpenSourceBrain/NEURONShowcase |
| neuroConstruct Showcase | Examples of neuroConstruct projects | https://github.com/OpenSourceBrain/neuroConstructShowcase |
| NeuroMorpho.Org | Examples of reconstructions from NeuroMorpho.Org | https://github.com/OpenSourceBrain/NeuroMorpho |
| Janelia MouseLight | Janelia MouseLight project neuronal reconstructions | https://github.com/OpenSourceBrain/MouseLightShowcase |

such as networks and neuronal models that reference multiple cell and ionic conductance definitions, can also be exported into a COMBINE zip archive (*Bergmann et al., 2014*), a zip file that includes metadata about its contents. pyNeuroML includes functions to easily create COMBINE archives from NeuroML models and simulations (*Figure 6*).

OSB is designed so that researchers can share their code on their chosen platform (e.g. GitHub), while retaining full control over write access to their repositories. Afterwards, a page for the model can be created on OSB which lists the latest files present there, with links to OSB visualization/analysis/simulation features which can use the standardized files found in the resource.

NeuroML supports the embedding of structured ontological information in model descriptions (*Neal et al., 2019*). Models can include NeuroLex (now InterLex) (*Larson and Martone, 2013*) identifiers for their components (e.g. neuro_lex_id in *Figure 6*). This links model components to their biological counterparts and makes them more transparent, findable, and reusable. For example, different types of neurons and brain regions have unique ontological ids. A user can use these ids to search for relevant model components on NeuroML-DB. More general information to maintain provenance can also be included in NeuroML models (https://docs.neuroml.org/Userdocs/Provenance.html).

## Reusing NeuroML models

NeuroML models, once openly shared, become community resources that are accessible to all. Researchers can use models shared on NeuroML-DB and OSB without restrictions. Guide 5 in *Table 5* provides an example of finding NeuroML-based model components using the API of NeuroML-DB, and creating novel models incorporating these elements.

In addition to these platforms, other experimental data and model dissemination platforms also provide standardized NeuroML versions of relevant models to promote uptake and reuse. For example, NeuroMorpho.org (*Ascoli et al., 2007*) includes a tool to download NeuroML compliant versions of its cellular reconstructions (https://github.com/NeuroML/Cvapp-NeuroMorpho.org, https://docs.neuroml.org/Userdocs/Software/Tools/SWC.html). NeuroML versions of models released by organizations such as the Blue Brain Project (*Markram et al., 2015*) (whole cell models as well as ion channel models from Channelpedia *Ranjan et al., 2011*), the Allen Institute for Brain Science (*Billeh et al., 2020*), and the OpenWorm project (*Gleeson et al., 2018*) are also openly available for reuse (*Table 8*).

NeuroML can also interact with other standards to further promote model re-use. Whereas NeuroML is a declarative standard, PyNN (*Davison et al., 2008*) is a procedural standard with a Python API for creating network models that can be simulated on multiple simulators. NeuroML models which are within the scope of PyNN can be converted to the PyNN format, and vice-versa. Similarly, NeuroML also interacts with SONATA (*Dai et al., 2020*) data format by supporting the two way conversion of the network structures of NeuroML models into SONATA. In standards not specific to neuroscience, models from the well established SBML standard (*Hucka et al., 2003*) can be converted to LEMS (*Cannon et al., 2014*), for inclusion in neuroscience-related modeling pipelines, and a subset of NeuroML/LEMS models can be exported to SBML, which allows use with simulators and analysis packages compliant to this standard, e.g., Tellurium (*Choi et al., 2018*). Simulation execution details of NeuroML/LEMS models can also be exported to Simulation Experiment Description Markup Language (SED-ML) (*Waltemath et al., 2011*), allowing advanced resources such as Biosimulators (*Shaikh et al., 2022*) (https://biosimulators.org) to feature NeuroML models.

## NeuroML is extensible

While the standard NeuroML elements (*Tables 1 and 2*) provide a broad range of curated model types for simulation-based investigations, NeuroML can be extended (using LEMS) to incorporate novel model elements and types when they are not (yet) available in the standard.

LEMS is a general purpose model specification language for creating fully machine readable definitions of the structure and behavior of model elements (*Cannon et al., 2014*). The dynamics of NeuroML elements are described in LEMS. The hierarchical nature of LEMS means that new elements can build on pre-existing elements of the modular NeuroML framework. For example, a novel ionic conductance element can extend the 'ionChannelHH' element, which in turn extends 'baseIonChannel.' Thus, the new element will be known to the NeuroML elements as depending on an external voltage and producing a conductance, properties that are inherited from 'baseIonChannel.' Other elements, such as a cell, can incorporate this new type without needing any other information about its internal workings.

LEMS (and, therefore, NeuroML) element definitions (called 'ComponentTypes') specify the dynamical behavior of the model element in terms of a list of yet to be set parameters. Once the generic model behavior is defined, modelers only need to fill in the appropriate values of the required parameters (e.g. conductance density, reversal potential, etc.) to create new instances (called 'Components') of the element (see Methods for more details). Users can, therefore create arbitrary, reusable model elements in LEMS, which can be treated the same way as the standard model elements (for an example see Guide 7 in *Table 5*).

Another major advantage of NeuroML's use of the LEMS language is its translatability. Since LEMS is fully machine readable, its primitives (e.g. state variables and their dynamics, expressed as ordinary differential equations) can be readily mapped into other languages. As a result, simulator specific code (*Blundell et al., 2018*) can be generated from NeuroML models and their LEMS extensions (*Figure 5*), allowing NeuroML to remain simulator-independent while supporting multiple simulation engines.

Newly created elements that may be of interest to the wider research community can be submitted to the NeuroML Editorial Board for inclusion into the standard. The standard, therefore, evolves as new model elements are added and improved versions of the standard and associated software tool chain are regularly released to the community.

## NeuroML is a global open community initiative

NeuroML is a global open community standard that is used and maintained collectively by a diverse set of stakeholders. The NeuroML Scientific Committee (https://docs.neuroml.org/NeuroMLOrg/ScientificCommittee.html) and the elected NeuroML Editorial Board (https://docs.neuroml.org/NeuroMLOrg/Board.html) oversee the standard, the core tools, and the initiative. This ensures that NeuroML supports the myriad of use cases generated by a multi-disciplinary computational modeling community.

NeuroML is an endorsed INCF (*Abrams et al., 2022*) community standard (*Martone and Das, 2019*) and is one of the main standards of the international COMBINE initiative (*Hucka et al., 2015*), which supports the development of other standards in computational biology as well (e.g. SBML (*Hucka et al., 2003*) and CellML *Lloyd et al., 2004*). Participation in these organizations guarantees that NeuroML follows current best practices in standardization, and remains linked to and interoperable with other standards wherever possible. The NeuroML community also participates in training and outreach activities such as Google Summer of Code (https://docs.neuroml.org/NeuroMLOrg/OutreachTraining.html), tutorials, and internships under these and other organizations.

The NeuroML community maintains public open communication channels to ensure that all community members can easily participate in troubleshooting, discussions, and development activities. A public mailing list (https://lists.sourceforge.net/lists/listinfo/neuroml-technology) is used for asynchronous communication and announcements while open chat channels on Gitter (now Matrix/Element (#/#NeuroML_community:gitter.im)) provide immediate access to the NeuroML community. All software repositories hosted on GitHub also have issue trackers for software specific queries. A community Code of Conduct (https://docs.neuroml.org/NeuroMLOrg/CoC.html) sets the standards of communication and behavior expected on all community channels.

A crucial aim of NeuroML is to enable Open Science and ensure models in computational neuroscience are FAIR. To this end, all development and discussions related to NeuroML are done publicly. The schema, all core software tools, and relevant resources such as documentation are made freely available under suitable Free/Open Source Software (FOSS) licenses on public platforms. Everyone can, therefore, use, modify, study, and share all NeuroML artifacts without restriction. Users and developers are encouraged to contribute modifications and improvements to the schema and core tools and to participate in the general maintenance and release process.

## Discussion

NeuroMLv2 has matured into a widely adopted community standard for computational neuroscience. Its modular, hierarchical structure can define a wide range of neuronal and circuit model types including simplified representations and those with a high degree of biological detail. The standardized, machine readable format of the NeuroMLv2/LEMS framework provides a flexible, common language for communicating between a wide range of tools and simulators used to create, validate, visualize, analyze, simulate, share, and reuse models. By enabling this interoperability, NeuroMLv2 has spawned a large ecosystem of interacting tools that cover all stages of the model development life cycle, bringing greater coherence to a previously fragmented landscape. Moreover, the modular nature of the model components and hierarchical structure conferred by NeuroMLv2, combined with the flexibility of coding in Python, has created a powerful 'building block' approach for constructing standardized models from scratch.

NeuroML has, therefore, evolved from a standardized archiving format into a mature language that supports an ecosystem of tools for the creation and execution of models that support the FAIR principles and promote open, transparent, and reproducible science.

### Evolution of NeuroML and emergence of the NeuroMLv2 tool ecosystem

NeuroML was conceived (*Goddard et al., 2001*) and developed (*Gleeson et al., 2010*) as a declarative XML-based framework for defining biophysical models of neurons and networks in a standardized form in order to compare model properties across simulators and to promote transparency and reuse. NeuroML version 1 achieved these aims and was mainly used to archive and visualize existing models (*Gleeson et al., 2010*). Building on this, the subsequent development of the NeuroMLv2/LEMS framework provided a way to describe models as a hierarchical set of components with dimensional parameters and state variables, so that their structure and dynamics are fully machine readable (*Cannon et al., 2014*). This enabled models to be losslessly mapped to other representations, greatly promoting interoperability between tools through read-write and automated code generation (*Blundell et al., 2018*). As NeuroMLv2 matured and became a community standard recognized by the INCF with a formal governance structure, an increasingly wide range of models and modeling tools have been developed or modified to be NeuroMLv2 compliant (*Tables 8, 3 and 4*). The core tools, maintained directly by the NeuroML developers (*Figure 4*), provide functionality to read, modify, or create new NeuroML models, as well as to analyze and visualize, and simulate the models. Furthermore, there are now a larger number of tools that have been developed by other members of the community (*Figure 3*) including a neuronal simulator designed specifically for NeuroMLv2 (*Panagiotou et al., 2022*). The emergence of an ecosystem of NeuroMLv2 compliant tools enables modelers to build tool chains that span the model life cycle and build and reuse standardized models.

### NeuroML and other standards in computational neuroscience

Several other standards and formats exist to support computational modeling of neuronal systems. Whereas NeuroML is a modular, declarative simulator independent standard for biophysical neuronal modeling, PyNN (*Davison et al., 2008*) and SONATA (*Dai et al., 2020*) provide a procedural Python-based simulator independent API and a framework for efficiently handling large-scale network simulations, respectively. Even though there is some overlap in the functionality provided by these standards, they each target distinct use cases and have their own goals and features. The teams developing these standards work in concert to ensure that they remain interoperable with each other, frequently sharing methods and techniques (*Dai et al., 2020*). This allows researchers to use their standard of choice and

easily combine with another if the need arises. PyNN and SONATA are, therefore, integral parts of the wider NeuroML ecosystem.

## Why using NeuroML and Python promotes the construction of FAIR models

The modular and hierarchical structure of NeuroMLv2, when combined with Python, provides a powerful combination of structured declarative elements and flexible procedural approaches that enables a 'Lego-like' building block approach for constructing biologically detailed models (*Cayco-Gajic et al., 2017*; *Billings et al., 2014*; *Kriener et al., 2022*; *Gurnani and Silver, 2021*). This has been advanced by the development of pyNeuroML, which provides a single installable package offering direct access to a range of functionality for handling NeuroML models (*Figure 6*). Moreover, the web-based documentation of NeuroMLv2, with multiple Python scripts illustrating the usage of the language and associated tools (*Table 5*), has recently been updated and expanded (https://docs.neuroml.org). This provides a central resource for both new and experienced users of NeuroML supporting its use in model building. As the examples of this resource illustrate, building models using NeuroMLv2 is efficient and intuitive, as the model components are pre-made and how they fit together specified. The structured format allows APIs like libNeuroML to incorporate features such as auto-completion and inline validation of model parameters and structure as scripts are being developed. In addition, automated multi-stage model validation ensures the code, equations and internal structure are validated against the NeuroML schema minimizing human errors and model simulations outputs are within acceptable bounds (*Figure 7*). The NeuroMLv2 ecosystem also provides convenient ways to visualize and inspect the inner structure of models. pyNeuroML provides Python functions and corresponding command line utilities to view neuronal morphology (*Figure 8*), neuronal electrophysiology (*Figure 10*), circuit connectivity and schematics (*Figure 9*). In addition, custom analysis pipelines and advanced neuroinformatics resources can easily be built using the APIs. For example, loading a NeuroML model of a neuron into OSB enables visualization of the morphology and the spatial distribution of ionic conductance over the membrane as well as inspection of the conductance state variables, while the connectivity and synaptic weight matrices can be automatically displayed for circuit models (*Figure 8*; *Gleeson et al., 2019b*). Such features of OSB, which are made possible by the structured format of NeuroMLv2, promote model transparency, reproducibility, and sharing. By enabling the development and sharing of well tested and transparent models the wider NeuroMLv2 ecosystem promotes Open Science.

## Limitations of NeuroML and current work

A limitation of any standardized framework is that there will always be models and model elements that fall outside the current scope of the standard. Although NeuroML suffers from this limitation, the underlying LEMS-based framework provides a flexible route to develop a wide range of new types of physio-chemical models (*Cannon et al., 2014*). This is relatively straightforward if the new model component, such as a synaptic plasticity mechanism, fits within the existing hierarchical structure of NeuroMLv2 as the new type of synaptic element can build on an existing base synapse type which specifies the relevant input and outputs (e.g. local voltage and synaptic current). For more radical shifts in model types (e.g. neuronal morphologies that grow during learning) that do not fit neatly into the current NeuroMLv2 schema, structural changes to the language would be required. This route is more involved as the pros and cons of changes to the structure of NeuroMLv2 would need to be considered by the Scientific Committee and, if approved, implemented by the Editorial Board.

Whereas the current scope of NeuroMLv2 encompasses models of spiking neurons and networks at different levels of biological detail, plans are in place to extend its scope to include more abstract, rate-based models of neuronal populations (e.g. see *Wilson and Cowan, 1972*; *Mejias et al., 2016* in *Table 8*). Additionally, work is under way to extend current support for SBML (*Hucka et al., 2003*) based descriptions of chemical signaling pathways (*Cannon et al., 2014*), to enable better biochemical descriptions of sub-cellular activity in neurons and synapses.

There is a growing interest in the field for the efficient generation and serialization of large-scale network models, containing numbers of neurons closer to their biological equivalents (*Markram et al., 2015*; *Billeh et al., 2020*; *Einevoll et al., 2019*). While a multitude of applications in the NeuroML ecosystem support large-scale model generation (e.g. NetPyNE, neuroConstruct, PyNN), the default

serialization of NeuroML (XML) is inefficient for reading/writing/storing such extensive descriptions. NeuroML does have an internal format for serializing in the binary format HDF5 (see Methods), but has also recently added support for export of models to the SONATA data format (*Dai et al., 2020*) allowing efficient serialization of large-scale models. Even though individual instances of large-scale models are useful, the ability to generate families of these for multiple simulation runs and more particularly a way to encapsulate, examine and reuse templates for network models, is also required. A prototype package, NeuroMLlite (https://github.com/NeuroML/NeuroMLlite), has been developed which allows these concise network templates to be described and multiple instances of networks to be generated, and facilitates interaction with simulation platforms and efficient serialization formats.

As discoveries and insights in neuroscience inform machine learning and visa versa, there is an increasing need to develop a common framework for describing both biological and artificial neural networks. Model Description Format (MDF) has been developed to address this (*Gleeson et al., 2023*). This initiative has developed a standardized format, along with a Python API, which allows the specification of artificial neural networks (e.g. Convolutional Neural Networks, Recurrent Neural Networks) and biological neurons using the same underlying entities. Support for mapping MDF to/from NeuroMLv2/LEMS has been included from the start. This work will enable deeper integration of computational neuroscience and 'brain-inspired' networks in Artificial Intelligence (AI).

## Conclusion and vision for the future

NeuroMLv2 is already a mature community standard that provides a framework for standardizing biologically detailed neuronal network models. By providing a stable, common framework defining the essential entities required for biologically detailed neuronal modeling, NeuroML has spawned an ecosystem of tools that span all stages of the model development life cycle. In the short term, we envision the functionality of NeuroML to expand further and for new online resources that encourage the construction of FAIR models using pyNeuroML to be taken up by the community. The NeuroML development team are also beginning to explore how to combine NeuroML-based circuit models with musculo-skeletal simulations to enable models of the neural control of behavior. In the longer term, developing seamless interfaces between NeuroML and other domain specific standards will enable the development of more holistic models of the neural control of body systems across a wide range of organisms, as well as greater exchange of models and insights between computational neuroscience and AI.

## Materials and methods

NeuroMLv2 is formally specified by the NeuroMLv2 XML schema, which defines the allowed structure of XML files which comply to the standard, and the LEMS ComponentType definitions, which define the internal state variables of the underlying elements, providing a machine-readable specification of the time evolution of model components. The specification is backed up by a suite of software tools that support the model life cycle and the accompanying usage and development documentation.

We illustrate the key parts of this framework using the HindmarshRose cell model (*Hindmarsh and Rose, 1984*; *Figure 11*), which as an abstract point neuron model, serves as an appropriate simple NeuroMLv2 ComponentType.

### The NeuroML XML Schema

We begin with the NeuroMLv2 standard. The standard consists of two parts, each serving different functions:

1. the NeuroMLv2 XML schema
2. corresponding LEMS component type definitions

The NeuroMLv2 schema is a language independent data model that constrains the structure of a NeuroMLv2 *model description*. The NeuroML schema is formally described as an XML Schema document (https://neuroml.org/schema/neuroml2) in the XML Schema Definition (XSD) format, a recommendation of the World Wide Web Consortium (W3C) (https://www.w3.org/TR/xmlschema-1/). An XML document that claims to conform to a particular schema can be *validated* against the schema. All NeuroMLv2 model descriptions can, therefore, be validated against the NeuroMLv2 schema.

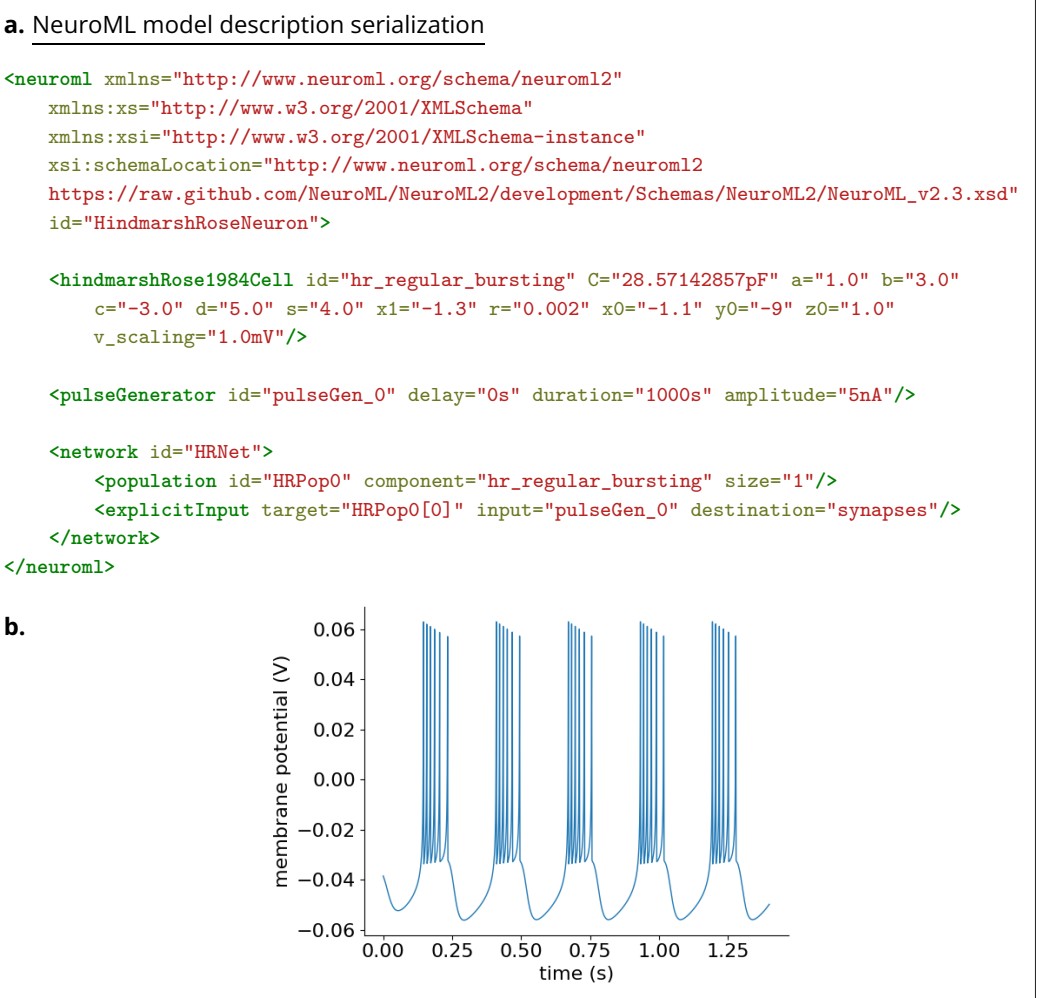

**a.** NeuroML model description serialization

```xml
<neuroml xmlns="http://www.neuroml.org/schema/neuroml2"
    xmlns:xs="http://www.w3.org/2001/XMLSchema"
    xmlns:xsi="http://www.w3.org/2001/XMLSchema-instance"
    xsi:schemaLocation="http://www.neuroml.org/schema/neuroml2
    https://raw.github.com/NeuroML/NeuroML2/development/Schemas/NeuroML2/NeuroML_v2.3.xsd"
    id="HindmarshRoseNeuron">

    <hindmarshRose1984Cell id="hr_regular_bursting" C="28.57142857pF" a="1.0" b="3.0"
        c="-3.0" d="5.0" s="4.0" x1="-1.3" r="0.002" x0="-1.1" y0="-9" z0="1.0"
        v_scaling="1.0mV"/>

    <pulseGenerator id="pulseGen_0" delay="0s" duration="1000s" amplitude="5nA"/>

    <network id="HRNet">
        <population id="HRPop0" component="hr_regular_bursting" size="1"/>
        <explicitInput target="HRPop0[0]" input="pulseGen_0" destination="synapses"/>
    </network>
</neuroml>
```

**b.**

**Figure 11.** Example model description of a HindmarshRose1984Cell NeuroML component. (**a**) XML serialization of the model description containing the main hindmarshRose1984Cell element with a set of parameters which result in regular bursting. A current clamp stimulus is applied using a pulseGenerator, and a population of one cell is added with this in a network. This XML can be validated against the NeuroML Schema. (**b**) Membrane potentials generated from a simulation of the model in (**a**). The LEMS simulation file to execute this is shown in *Figure 15*. The code used in this example is available here: https://github.com/OpenSourceBrain/HindmarshRose1984/tree/master/NeuroML2/examples.

The basic building blocks of an XSD schema are 'simple' or 'complex' types and their 'attributes.' All types are created as 'extensions' or 'restrictions' of other types. Complex types may contain other types and attributes whereas simple types may not. *Figure 12* shows some example types defined in the NeuroMLv2 schema. For example, the `Nml2Quantity_none` simple type restricts the in-built 'string' type using a regular expression 'pattern' that limits what string values it can contain. The type is `Nml2Quantity_none` is to be used for unit-less quantities (e.g. 3, 6.7, –1.1e-5) and the restriction pattern for translates to 'a string that may start with a hyphen (negative sign), followed by any number of numerical characters (potentially containing a decimal point) and a string containing capital or small 'e' (to specify the exponent).' The restriction pattern for the `Nml2Quantity_voltage` type is similar, but must be followed by a 'V' or 'mV.' In this way, the restriction ensures that a value of type 'Nml2Quantity_voltage' represents a physical voltage quantity with units 'V' (volt) or 'mV' (millivolt). Furthermore, a NeuroMLv2 model description that uses a voltage value that does not match this pattern, for example '0.5 s,' will be invalid.

The example of a complex type in *Figure 12* is the `HindmarshRose1984Cell` type that extends the `BaseCellMembPotCap` complex type (the base type for any cell producing a membrane potential

```
<xs:simpleType name="Nml2Quantity_none">  <!-- For dimensionless parameters -->
  <xs:restriction base="xs:string">
    <xs:pattern value="-?([0-9]*(\.[0-9]+)?)([eE]-?[0-9]+)?"/>
  </xs:restriction>
</xs:simpleType>

<xs:simpleType name="Nml2Quantity_voltage">  <!-- For params with dimension voltage -->
  <xs:restriction base="xs:string">
    <xs:pattern value="-?([0-9]*(\.[0-9]+)?)([eE]-?[0-9]+)?[\s]*(V|mV)"/>
  </xs:restriction>
</xs:simpleType>

<xs:complexType name="HindmarshRose1984Cell">
  <xs:annotation>
    <xs:documentation>The Hindmarsh Rose model is a simplified point cell model which
        captures complex firing patterns of single neurons, such as
        periodic and chaotic bursting...
    </xs:documentation>
  </xs:annotation>
  <xs:complexContent>
    <xs:extension base="BaseCellMembPotCap">
      <xs:attribute name="a" type="Nml2Quantity_none" use="required"/>
      <xs:attribute name="b" type="Nml2Quantity_none" use="required"/>
      <xs:attribute name="c" type="Nml2Quantity_none" use="required"/>
      <xs:attribute name="d" type="Nml2Quantity_none" use="required"/>
      <xs:attribute name="s" type="Nml2Quantity_none" use="required"/>
      <xs:attribute name="x1" type="Nml2Quantity_none" use="required"/>
      <xs:attribute name="r" type="Nml2Quantity_none" use="required"/>
      <xs:attribute name="x0" type="Nml2Quantity_none" use="required"/>
      <xs:attribute name="y0" type="Nml2Quantity_none" use="required"/>
      <xs:attribute name="z0" type="Nml2Quantity_none" use="required"/>
      <xs:attribute name="v_scaling" type="Nml2Quantity_voltage" use="required"/>
    </xs:extension>
  </xs:complexContent>
</xs:complexType>
```

**Figure 12.** Type definitions taken from the NeuroMLv2 schema (https://github.com/NeuroML/NeuroML2/blob/master/Schemas/NeuroML2/NeuroML_v2.3.1.xsd) which describes the structure of NeuroMLv2 elements. Top: 'simple' types may not include other elements or attributes. Here, the `Nml2Quantity_none` and `Nml2Quantity_voltage` types define restrictions on the default string type to limit what strings can be used as valid values for attributes of these types. Bottom: example of a 'complex' type, the HindmarshRose cell model (**Hindmarsh and Rose, 1984**), that can also include other elements of other types, and extend other types.

`v` with a capacitance parameter `C`), and defines new 'required' (compulsory) attributes. These attributes are of simple types—these are all unit-less quantities apart from `v_scaling`, which has dimension voltage. Note that inherited attributes are not re-listed in the complex type definition—the compulsory capacitance attribute, `C`, is inherited here from `BaseCellMembPotCap`.

The NeuroMLv2 schema serves multiple critical functions. A variety of tools and libraries support the validation of files against XSD schema definitions. Therefore, the NeuroMLv2 schema enables the validation of model descriptions—model structure, parameters, parameter values and their units, cardinality, element positioning in the model hierarchy (level 1 validation in *Figure 7*)—*prior to simulation*. XSD schema definitions, as language independent data models, also allow the generation of APIs in different languages. More information on how APIs in different languages are generated using the NeuroMLv2 XSD schema definition is provided in later sections.

The NeuroMLv2 XSD schema is also released and maintained as a versioned artifact, similar to the software packages. The current version is 2.3, and can be found in the NeuroML2 repository on GitHub (https://github.com/NeuroML/NeuroML2/tree/master/Schemas/NeuroML2).

## LEMS ComponentType definitions

The second part of the NeuroMLv2 standard consists of the corresponding LEMS ComponentType definitions. Whereas the XSD Schema describes the *structure* of a NeuroMLv2 model description, the LEMS `ComponentType` definitions formally describe the *dynamics* of the model elements.

LEMS (*Cannon et al., 2014*) is a domain independent general purpose machine-readable language for describing models and their simulations. A complete description of LEMS is provided in *Cannon et al., 2014* and in our documentation (https://docs.neuroml.org/Userdocs/LEMSSchema.html). Here, we limit ourselves to a short summary necessary for understanding the NeuroMLv2 `ComponentType` definitions.

LEMS allows the definition of new model types called `ComponentTypes`. These are formal descriptions of how a generic model element of that type behaves (the 'dynamics'), *independent of the specific set of parameters in any instance*. To describe the dynamics, such descriptions must list any necessary parameters that are required, as well as the time-varying state variables. The dimensions of these parameters and state variables must be specified, and any expressions involving them must be dimensionally consistent. An instance of such a generic model is termed a `Component` and can be instantiated from a `ComponentType` by providing the necessary parameters. One can think of `ComponentTypes` as user defined data types similar to 'classes' in many programming languages and Components as 'objects' of these types with particular sets of parameters. Types in LEMS can also extend other types, enabling the construction of a hierarchical library of types. In addition, since LEMS is designed for model simulation, `ComponentType` definitions also include other simulation-related features such as `Exposures`, specifying quantities that may be accessed/recorded by users.

For model elements included in the NeuroML standard, there is a one-to-one mapping between types specified in the NeuroML XSD schema and LEMS `ComponentTypes`, with the same parameters specified in each. The addition of new model elements to the NeuroML standard, therefore, requires the addition of new type definitions to both the XSD schema and the LEMS definitions. New user defined `ComponentTypes`, nevertheless, can be defined in LEMS and used freely in models, and these do not need to be added to the standard before use. The only limitation here is that new user defined `ComponentTypes` cannot be validated against the NeuroML schema since their type definitions will not be included there.

*Figure 13* shows the `ComponentType` definition for the `HindmarshRose1984Cell` model element. Here, the `HindmarshRose1984Cell` `ComponentType` extends `baseCellMembPotCap` and inherits its elements. The `ComponentType` includes parameters that users must provide when creating a new instance (component): $a, b, c, d, r, v, x1, v\_scaling$.

Other parameters, $x0, y0$, and $z0$ are used to initialize the three state variables of the model, $x, y, z$. $x$ is the proxy for the membrane potential of the cell used in the original formulation of the model (*Hindmarsh and Rose, 1984*) and is here scaled by a factor $v\_scaled$ to expose a more physiological value for the membrane potential of the cell in `StateVariable` $v$. A `Constant`, MSEC, is defined to hold the value of 1 ms for use in the `ComponentType`. Next, an `Attachment` enables the addition of entities that would provide external inputs to the `ComponentType`. Here, synapses are `Attachments` of the type `basePointCurrent` and provide synaptic current input to this `ComponentType`.

The `Dynamics` block lists the mathematical formalism required to simulate the `ComponentType`. By default, variables defined in the `Dynamics` block are private, i.e., they are not visible outside the `ComponentType`. To make these visible to other `ComponentTypes` and to allow users to record them, they must be connected to `Exposures`. Exposures for this `ComponentType` include the three state variables and also the internal derived variables, which while not used by other components, are useful in inspecting the `ComponentType` and its dynamics. An extra exposure, *spiking*, is added to allow other NeuroML components access to the spiking state of the cell that will be determined in the `Dynamics` block.

`StateVariable` definitions are followed by `DerivedVariables`, variables whose values depend on other variables but are not time derivatives (which are handled separately in `TimeDerivative` blocks (below)). The total synaptic current, *iSyn*, is a summation of all the synaptic currents, *i* received by the synapses that may be attached on to this `ComponentType`. The `synapse[*]/i` value of the select field tells LEMS to collect all the `i` exposures from any synapses `Attachments`, and the `add` value of the reduce field tells LEMS to sum the multiple values. As noted, $x$ is a scaled version of the membrane potential variable, $v$. This is followed by the three derived variables, *phi*, *chi*, *rho* where:

$$phi = y - ax^3 + bx^2 \tag{1}$$

$$chi = c - dx^2 - y \tag{2}$$

```xml
<ComponentType name="hindmarshRose1984Cell" extends="baseCellMembPotCap" description="The Hindmarsh Rose
  model">
  
  
  
  
  
  
  
  

  <!-- Initial Conditions -->
  
  
  

  <Constant name="MSEC" dimension="time" value="1ms"/>

  <Attachments name="synapses" type="basePointCurrent"/>

  <Exposure name="x" dimension="none"/>
  <Exposure name="y" dimension="none"/>
  <Exposure name="z" dimension="none"/>
  <Exposure name="phi" dimension="none"/>
  <Exposure name="chi" dimension="none"/>
  <Exposure name="rho" dimension="none"/>
  <Exposure name="spiking" dimension="none"/>
  <Dynamics>
    <StateVariable name="v" dimension="voltage" exposure="v"/>
    <StateVariable name="y" dimension="none" exposure="y"/>
    <StateVariable name="z" dimension="none" exposure="z"/>
    <StateVariable name="spiking" dimension="none" exposure="spiking"/>

    <DerivedVariable name="iSyn" dimension="current" exposure="iSyn" select="synapses[*]/i" reduce="add" />
    <DerivedVariable name="x" dimension="none" exposure="x" value="v / v_scaling"/>
    <DerivedVariable name="phi" dimension="none" exposure="phi" value="y - a * x^3 + b * x^2"/>
    <DerivedVariable name="chi" dimension="none" exposure="chi" value="c - d * x^2 - y"/>
    <DerivedVariable name="rho" dimension="none" exposure="rho" value="s * ( x - x1 ) - z"/>
    <DerivedVariable name="iMemb" dimension="current" exposure="iMemb"
                     value="(C * (v_scaling * (phi - z) / MSEC)) + iSyn"/>

    <TimeDerivative variable="v" value="iMemb/C"/>
    <TimeDerivative variable="y" value="chi / MSEC"/>
    <TimeDerivative variable="z" value="r * rho / MSEC"/>

    <OnStart>
      <StateAssignment variable="v" value="x0 * v_scaling"/>
      <StateAssignment variable="y" value="y0"/>
      <StateAssignment variable="z" value="z0"/>
    </OnStart>
    <OnCondition test="v .gt. 0 .and. spiking .lt. 0.5">
      <StateAssignment variable="spiking" value="1"/>
      <EventOut port="spike"/>
    </OnCondition>
    <OnCondition test="v .lt. 0">
      <StateAssignment variable="spiking" value="0"/>
    </OnCondition>
  </Dynamics>
</ComponentType>
```

**Figure 13.** LEMS `ComponentType` definition of the HindmarshRose cell model (*Hindmarsh and Rose, 1984*, https://github.com/NeuroML/NeuroML2/blob/master/NeuroML2CoreTypes/Cells.xml).

$$rho = s(x - x1) - z \qquad (3)$$

The total membrane potential of the cell, *iMemb*, is calculated as the sum of the capacitive current and the synaptic current:

$$iMemb = \frac{C(v\_scaling(phi - z))}{\text{MSEC}} + iSyn \qquad (4)$$

$v, y, z$ are TimeDerivatives, with the 'value' representing the rate of change of each variable:

$$dv/dt = iMemb/C \tag{5}$$
$$dy/dt = chi/\text{MSEC} \tag{6}$$
$$dz/dt = (r \times rho)/\text{MSEC} \tag{7}$$

The final few blocks set the initial state of the component (OnStart),

$$v = x0 \times v\_scaling \tag{8}$$
$$y = y0 \tag{9}$$
$$z = z0 \tag{10}$$

and define conditional expressions to set the spiking state of the cell:

$$spiking = \begin{cases} 1 & \text{if } (v > 0) \wedge (spiking < 0.5) \\ 0 & \text{if } (v < 0) \end{cases} \tag{11}$$

Both the XSD schema and the LEMS `ComponentType` definitions enable model validation. However, despite some overlap, they support different types of validation. Whereas the XSD schema allows for the validation of *model descriptions* (e.g. the XML files), the LEMS `ComponentType` definitions enable validation of *model instances*, i.e., the 'runnable' instances of models that are constructed once components have been created by instantiating `ComponentTypes` with the necessary parameters, and various attachments created between source and target components. A model description may be used to create many different model instances for simulation. Indeed, it is common practice to run models that include stochasticity with different seeds for random number generators to verify the robustness of simulation results. Thus, the validation of dimensions and units that LEMS carries out is done only after a runnable instance of a model has been created.

The LEMS `ComponentType` definitions for NeuroMLv2 are also maintained as versioned files that are updated along with the XSD schema. These can also be seen in the NeuroMLv2 GitHub repository (https://github.com/NeuroML/NeuroML2/tree/master/NeuroML2CoreTypes). An index of the `ComponentTypes` included in version 2.3 of the NeuroML standard, with links to online documentation, is also provided in *Tables 1 and 2*.

## NeuroML APIs

The NeuroMLv2 software stack relies on the NeuroML APIs that provide functionality to read, write, validate, and inspect NeuroML models. The APIs are programmatically generated from the machine readable XSD schema, thus ensuring that the class for defining a specific NeuroML element in a given language (e.g. Java) has the correct set of fields with the appropriate type (e.g. float or string) corresponding to the allowed parameters in the corresponding NeuroML element. NeuroMLv2 currently provides APIs in numerous languages—Python (libNeuroML which is generated via generateDS (http://www.davekuhlman.org/generateDS.html)), Java (org.neuroml.model via JAXB XJC (https://javaee.github.io/jaxb-v2/)), C++ (NeuroML_CPP via XSD (https://www.codesynthesis.com/products/xsd/)) and MATLAB (NeuroMLToolbox which accesses the Java API from MATLAB), and APIs for other languages can also be easily generated as required. LEMS is also supported by a similar set of APIs—PyLEMS in Python, and jLEMS in Java—and since a NeuroMLv2 model description is a set of LEMS `Components`, the LEMS APIs also support them (e.g. the `hindmarshRose1984`Cell example in *Figure 11* could be loaded by jLEMS and treated as a LEMS `Component`).

*Figure 14* shows the use of the NeuroML Python API to describe a model with one HindmarshRose cell. In Python, the instances of `ComponentTypes`, their `Components`, are represented as Python objects. The `hr0` Python variable stores the created `HindmarshRose1984`Cell component/object. This is added to a `Population pop0` in the `Network net`. The network also includes a `PulseGenerator` with amplitude 5 nA as an `ExplicitInput` that is targeted at the cell in the population. The model description is serialized to XML (*Figure 11*) and validated. Note that as the standard convention for classes in Python is to use capitalized names, `HindmarshRose1984Cell` is used in Python but is serialized as `<hindmarshRose1984Cell>`in the XML. Users can either share the Python script

> Create a new HindmarshRose cell component with parameters for regular spiking
>
> ```python
> nml_doc = component_factory("NeuroMLDocument", id="HindmarshRoseNeuron")
> hr0 = nml_doc.add("HindmarshRose1984Cell", id="hr_regular", a="1.0", b="3.0", c="-3.0", d="5.0",
>     s="4.0", x1="-1.3", r="0.002", x0="-1.1", y0="-9", z0="1.0", C="28.57142857pF",
>     v_scaling="35.0mV")
> net = nml_doc.add("Network", id="HRNet", validate=False)
> ```
>
> Create a population of cells (1 cell)
>
> ```python
> pop0 = net.add("Population", id="HRPop0", component=hr0.id, size=1)
> ```
>
> Add external stimuli to the population
>
> ```python
> pg = nml_doc.add("PulseGenerator", id="pulseGen_%i" % 0, delay="0s", duration="1000s",
>     amplitude="5nA")
> exp_input = net.add("ExplicitInput", target="%s[%i]" % (pop0.id, 0), input=pg.id,
>     destination="synapses")
> ```
>
> Save (serialize) the model to a file
>
> ```python
> nml_file = 'hindmarshrose1984_single_cell_network.nml'
> writers.NeuroMLWriter.write(nml_doc, nml_file)
> ```
>
> Validate the model
>
> ```python
> validate_neuroml2(nml_file)
> ```

**Figure 14.** Example model description of a HindmarshRose1984Cell NeuroML component in Python using parameters for regular bursting. This script generates the XML in **Figure 11**. The code used in this example is available here: https://github.com/OpenSourceBrain/HindmarshRose1984/tree/master/NeuroML2/examples.

itself or share the XML serialization. Any valid XML serialization can be also loaded into a Python object model and modified.

XML is the default serialization of NeuroML and all existing APIs can read and write the format (and it should be seen as a minimal requirement for new APIs to support XML). There is, however, an alternative HDF5 (https://www.hdfgroup.org/solutions/hdf5) based serialization of NeuroML files which is supported by both libNeuroML and the Java API, org.neuroml.model (https://docs.neuroml.org/User-docs/HDF5.html). This format is based on an efficient representation of cell positions and connectivity data as HDF5 data sets which can be serialized in compact binary format and loaded into memory for optimized access (e.g. as numpy arrays in libNeuroML). This reduces the size of the saved files for large-scale networks and speeds up loading/writing models eliminating the need to parse/generate large text files containing XML. Models serialized in this format can be loaded and transformed to simulator code in the same way as XML-based models by the Java and Python APIs.

## Simulating NeuroML models

The model description shown in **Figure 11** contains no information about how it is to be simulated, or on the dynamics of each model component. Providing this simulation information and linking in the `ComponentType` definition requires creating a LEMS file to fully specify the simulation. **Figure 15** shows the use of utilities included in the Python pyNeuroML package to describe a LEMS simulation of the HindmarshRose model defined in **Figure 11**. The `LEMSSimulation` object includes simulation specific information such as the duration of the simulation, the integration time step, and the seed value. It also allows the specification of files for the storage of data recorded from the simulation. In this example, we record the membrane potential, $v$, of our cell in its population, `HRPop0[0]`. Similar to the NeuroMLv2 model description, the simulation object can also be serialized to XML for storage and sharing (**Figure 15**, bottom).

As noted previously, NeuroML/LEMS model and simulation descriptions are machine readable and simulator independent and can be simulated by simulation engines using a multitude of strategies (**Figure 5**).

The first category of tools consists of the reference NeuroML and LEMS simulation engines. These work directly with NeuroML and LEMS as their base descriptions of modeling entities and do not

<div style="border:1px solid;padding:1em">

Create a simulation of the model

```
simulation_id = "example-single-hindmarshrose1984cell-sim"
simulation = LEMSSimulation(sim_id=simulation_id, duration=1.4e3, dt=0.0025, simulation_seed=123)
simulation.assign_simulation_target(net.id)
simulation.include_neuroml2_file(nml_file)
```

Record membrane potential to an output file

```
simulation.create_output_file("output0", "%s.v.dat" % simulation_id)
simulation.add_column_to_output_file("output0", 'HRPop0[0]', 'HRPop0[0]/v')
```

Save the simulation to file and run it in jNeuroML/jLEMS

```
lems_simulation_file = simulation.save_to_file()
pynml.run_lems_with_jneuroml(lems_simulation_file, max_memory="2G", nogui=True, plot=False)
```

LEMS simulation description serialization

```xml
<Lems>
    <!-- Specify which component to run -->
    <Target component="example-single-hindmarshrose1984cell-sim"/>

    <!-- Include core NeuroML2 ComponentType definitions -->
    <Include file="Cells.xml"/>
    <Include file="Networks.xml"/>
    <Include file="Simulation.xml"/>

    <Include file="hindmarshrose1984_single_cell_network.nml"/>

    <Simulation id="example-single-hindmarshrose1984cell-sim" length="1400.0ms" step="0.0025ms"
        target="HRNet" seed="123">  <!-- Note seed: ensures same random numbers used every run
        -->
        <OutputFile id="output0" fileName="example-single-hindmarshrose1984cell-sim.v.dat">
            <OutputColumn id="HRPop0[0]" quantity="HRPop0[0]/v"/>
        </OutputFile>
    </Simulation>
</Lems>
```

</div>

**Figure 15.** An example simulation of the HindmarshRose model description shown in *Figure 14* with the LEMS serialization shown at the bottom. The code used in this example is available here: https://github.com/OpenSourceBrain/HindmarshRose1984/tree/master/NeuroML2/examples.

have their own specific formats. They are maintained by the NeuroML Editorial Board—jLEMS, jNeuroML, and PyLEMS (*Figure 4*). jLEMS serves as the reference implementation for the LEMS language and as such it can simulate any model described in LEMS (not necessarily from neuroscience). When coupled with the LEMS definitions of NeuroML standard entity structure/dynamics, it can simulate most NeuroML models, though it does not currently support multi-compartmental neurons. jNeuroML bundles the NeuroML standard LEMS definitions, jLEMS, and other functionality into a single package for ease of installation/usage. There is also a pure Python implementation of a LEMS interpreter, PyLEMS, which can be used in a similar way to jLEMS. The pyNeuroML package encapsulates all of these tools to give easy access (at both command line and in Python) to all of their functionality (*Figure 6*).

The second category consists of other simulators which support NeuroML natively. The EDEN simulator is an independently developed tool that was designed from its inception to read NeuroML and LEMS models for efficient, parallel simulation (*Panagiotou et al., 2022*).

The third category involves simulators which have their own internal formats and include methods to translate NeuroMLv2/LEMS models to their own formats. Examples include NetPyNE (*Dura-Bernal et al., 2019*), MOOSE (*Ray and Bhalla, 2008*), and N2A (*Rothganger et al., 2014*).

The fourth category comprises tools for which the NeuroML tools generate simulator specific scripts. The simulation engines then execute these scripts, similar to how they would execute handwritten

user scripts. These include NEURON (*Hines and Carnevale, 1997*) for which the NeuroML tools generate scripts in Python and the simulator's hoc and NMODL formats and the Brian simulator (*Stimberg et al., 2019*) which uses Python scripts.

The final category consists of export options to standardized formats in neuroscience and the wider computational biology field, which enable interaction with simulators and applications supporting those formats. These include the PyNN package (*Davison et al., 2008*), which can be run in either NEURON, NEST (*Gewaltig and Diesmann, 2007*) or Brian, the SONATA data format (*Dai et al., 2020*) and the SBML standard (*Hucka et al., 2003*) (see Reusing NeuroML models for more details).

Having multiple strategies in place for supporting NeuroML gives more freedom to simulator developers to choose how much they wish to be involved with implementing and supporting NeuroML functionality in their applications, while maximizing the options available for end users.

The primary tool for simulating NeuroML/LEMS models via different engines is jNeuroML, which is included in pyNeuroML. jNeuroML supports all simulator engine categories (*Figure 5*). It includes jLEMS for simulation of LEMS and single compartmental NeuroML models. It can also pass simulations to the EDEN simulator (*Panagiotou et al., 2022*) for direct simulation. Using the org.neuroml. export library (https://github.com/NeuroML/org.neuroml.export), jNeuroML can also generate import scripts for simulators (e.g. NetPyNE *Dura-Bernal et al., 2019*) or convert NeuroML/LEMS models to simulator specific formats (e.g. NEURON *Hines and Carnevale, 1997*). Supporting a new simulation engine that requires translation of NeuroML/LEMS into another format can be done by adding a new 'writer' to the org.neuroml.export library. Finally, jNeuroML also includes the org.neuroml.import (https://github.com/NeuroML/jNeuroML) library that converts from other formats (e.g. SBML *Hucka et al., 2003*) to LEMS for combination with NeuroML models.

It is important to note though that not all NeuroML models can be exported to/are supported by each of these target simulators (*Table 7*). This depends on the capabilities of the simulator in question (whether it supports networks, or morphologically detailed cells) and pyNeuroML/jNeuroML will provide feedback if a feature of the model is not supported in a chosen environment.

All NeuroML and LEMS software packages are made available under FOSS licenses. The source code for all NeuroML packages and the standard can be obtained from the NeuroML GitHub organization (https://github.com/NeuroML). The NeuroML Python API (https://github.com/NeuralEnsemble/libNeuroML) was developed in collaboration with the NeuralEnsemble initiative (https://github.com/NeuralEnsemble/), which also maintains other commonly used Python packages such as PyNN (*Davison et al., 2008*), Neo (*Garcia et al., 2014*) and Elephant (*Denker, 2018*). LEMS packages are available from the LEMS GitHub organization (https://github.com/LEMS).

To ensure replication and reproduction of studies, it is important to note the exact versions of software used in studies. For NeuroML and LEMS packages, archives of each release along with citations are published on Zenodo (https://zenodo.org) to enable researchers to cite them in their work (*Gleeson, 2021*; *Gleeson, 2024a*; *Gleeson et al., 2019b*; *Gleeson, 2024b*; *Sinha, 2024*).

## Documentation

A standard and its accompanying software ecosystem must be supported by comprehensive documentation if it is to be of use to the research community. The primary NeuroML documentation for users that accompanies this paper has been consolidated into a JupyterBook (*Executable Books Community, 2020*) at https://docs.neuroml.org. This includes explanations of NeuroML and computational modeling concepts, interactive tutorials with varying levels of complexity, information about tools and what functions they provide to support different stages of the model life cycle. The JupyterBook framework supports 'executable' documentation through the inclusion of interactive Jupyter notebooks which may be run in the users' web browser on free services such as OSBv2, Binder.org (https://mybinder.org/) and Google Colab (https://colab.research.google.com/). Finally, the machine readable nature of the schema and LEMS also enables the automated generation of human readable documentation for the standard and low level APIs (*Figure 16*) along with their examples (https://docs.neuroml.org/Userdocs/Schemas/Cells.html#hindmarshrose1984cell). In addition, the individual NeuroML software packages each have their own individual documentation (e.g. pyNeuroML (https://pyneuroml.readthedocs.io/en/stable/,) libNeuroML (https://libneuroml.readthedocs.io/en/stable/)).

As with the rest of the NeuroML ecosystem, the documentation is hosted on GitHub (https://github.com/NeuroML/Documentation), licensed under a FOSS license, and community contributions

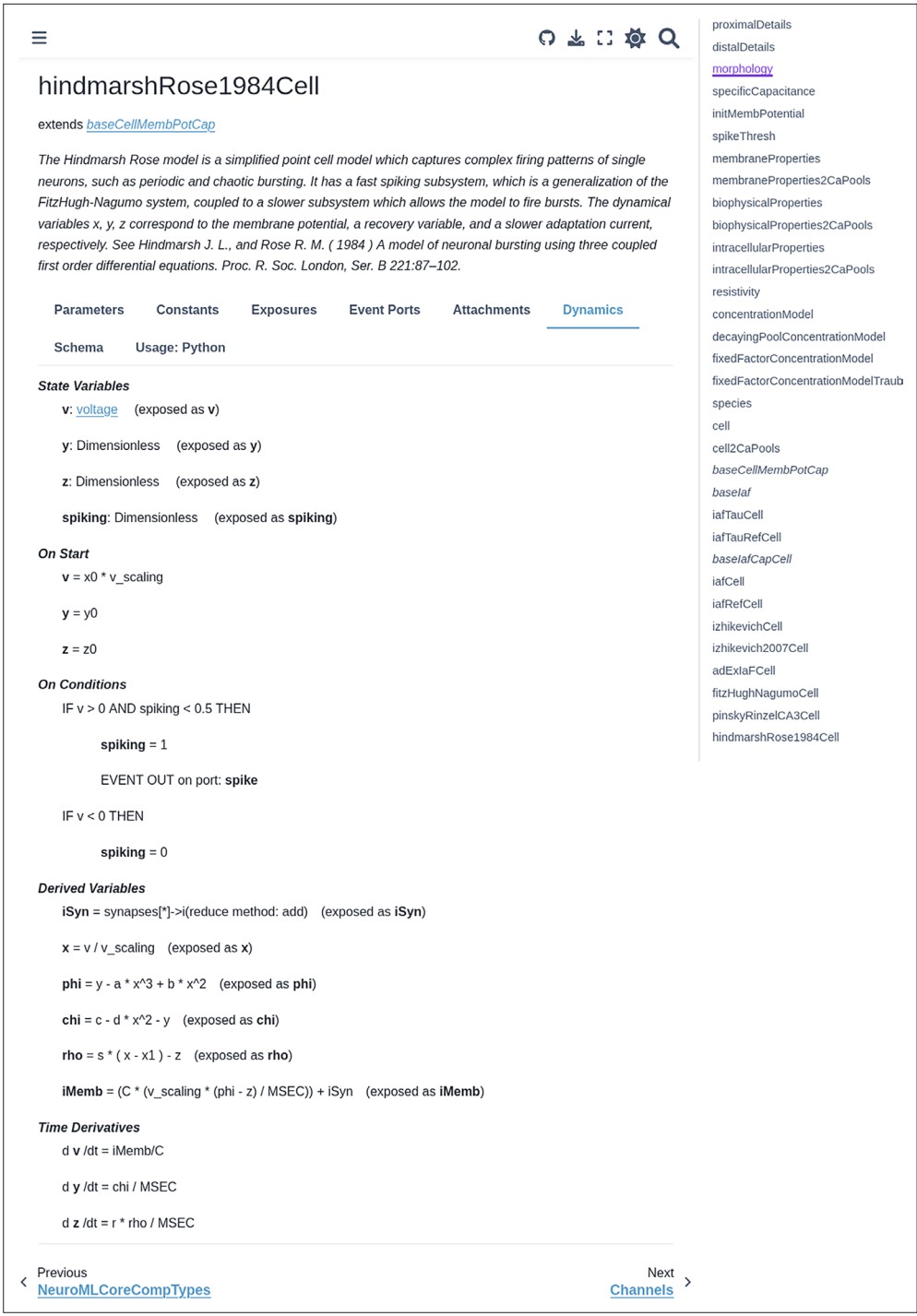

**Figure 16.** Documentation for the `HindmarshRose1984Cell` NeuroMLv2 `ComponentType` generated from the XSD schema and LEMS definitions on the NeuroML documentation website showing its dynamics (https://docs.neuroml.org/Userdocs/Schemas/Cells.html#hindmarshrose1984cell). More information about the ComponentType can be obtained from the tabs provided.

to it are welcomed. A PDF version of the documentation can also be downloaded for offline use (https://docs.neuroml.org/_static/files/neuroml-documentation.pdf).

## Maintenance of the Schema and core software

The NeuroML Scientific Committee (https://docs.neuroml.org/NeuroMLOrg/ScientificCommittee.html) and the elected NeuroML Editorial Board (https://docs.neuroml.org/NeuroMLOrg/Board.html) oversee the standard, the core tools, and the initiative. The Scientific Committee sets the scientific focus of the NeuroML initiative. It ensures that the standard represents the state of the art—that it can encapsulate the latest knowledge in neuronal anatomy and physiology in their corresponding model components. The Scientific Committee also defines the governance structure of the initiative and works with the wider scientific community to gather feedback on NeuroML and promote its use. The Editorial Board manages the day-to-day development and maintenance of LEMS, the NeuroML schema, the core software tools, and critical resources such as the documentation. The Editorial Board works with simulator developers in the extended ecosystem to help make tools NeuroML compliant by testing reference implementations and answering technical queries about NeuroML and the core software tools.

## Acknowledgements

We thank all the members of the NeuroML Community who have contributed to the development of the standard over the years, have added support for the language to their applications, or who have converted published models to NeuroML. We would particularly like to thank the following for contributions to the NeuroML Scientific Committee: Upi Bhalla, Avrama Blackwell, Hugo Cornells, Robert McDougal, Lyle Graham, Cengiz Gunay, and Michael Hines. The following have also contributed to developments related to the named tools/simulators/resources: EDEN - Mario Negrello and Christos Strydis, SONATA - Anton Arkhipov and Kael Dai, MOOSE - Subhasis Ray, NeuroML-DB - Justas Birgiolas, NeuroMorpho.Org - Giorgio Ascoli, N2A - Fred Rothganger, pyLEMS - Gautham Ganapathy, MDF - Manifest Chakalov, libNeuroML and NeuroTune - Mike Vella, Open Source Brain - Matt Earnshaw, Adrian Quintana and Eugenio Piasini, SciUnit/NeuronUnit - Richard C Gerkin, Brian - Marcel Stimberg and Dominik Krzemiński, Arbor - Nora Abi Akar, Thorsten Hater and Brent Huisman, BluePyOpt - Jaquier Aurélien Tristan and Werner van Geit, C++/MATLAB APIs - Jonathan Cooper. We thank Rokas Stanislavos, András Ecker, Jessica Dafflon, Ronaldo Nunes, Anuja Negi, and Shayan Shafquat for their work converting models to NeuroML format as part of the Google Summer of Code program. We also thank Diccon Coyle for feedback on the manuscript.

## Additional information

### Competing interests

Matteo Cantarelli: MetaCell Ltd. was contracted by UCL to develop some of the NeuroML support on the Open Source Brain platform; MC has a financial interest in MetaCell Ltd. Robert C Cannon: Employee of Opus2 International Ltd. The other authors declare that no competing interests exist.

### Funding

| Funder | Grant reference number | Author |
| --- | --- | --- |
| Wellcome Trust | 10.35802/101445 | Padraig Gleeson<br>Robin Angus Silver |
| Wellcome Trust | 10.35802/212941 | Padraig Gleeson<br>Robin Angus Silver |
| Wellcome Trust | 10.35802/203048 | Robin Angus Silver |
| Wellcome Trust | 10.35802/224499 | Robin Angus Silver |
| Kavli Foundation | LS-2022-GR-40-2648 | Padraig Gleeson |

| Funder | Grant reference number | Author |
|---|---|---|
| Engineering and Physical Sciences Research Council | EP/X011151/1 | Padraig Gleeson |
| National Institutes of Health | MH081905 | Sharon Crook |
| National Institutes of Health | EB014640 | Sharon Crook |
| National Institutes of Health | MH106674 | Sharon Crook |
| National Institutes of Health | U24EB028998 | Salvador Dura-Bernal |
| New York State Department of Health - Wadsworth Center | DOH01-C38328GG | Salvador Dura-Bernal |
| HORIZON EUROPE Framework Programme | SEPTON (Gr. Agr. No. 101094901) | Sotirios Panagiotou |

The funders had no role in study design, data collection and interpretation, or the decision to submit the work for publication. For the purpose of Open Access, the authors have applied a CC BY public copyright license to any Author Accepted Manuscript version arising from this submission.

## Author contributions

Ankur Sinha, Conceptualization, Resources, Data curation, Software, Formal analysis, Validation, Investigation, Visualization, Methodology, Writing – original draft, Project administration, Writing – review and editing; Padraig Gleeson, Conceptualization, Resources, Data curation, Software, Formal analysis, Supervision, Funding acquisition, Validation, Investigation, Visualization, Methodology, Writing – original draft, Project administration, Writing – review and editing; Bóris Marin, Conceptualization, Resources, Software, Validation, Investigation, Methodology, Writing – review and editing; Salvador Dura-Bernal, Conceptualization, Resources, Software, Funding acquisition, Investigation, Visualization, Methodology, Writing – review and editing; Sotirios Panagiotou, Resources, Software, Validation, Investigation, Methodology, Writing – review and editing; Sharon Crook, Conceptualization, Resources, Software, Supervision, Funding acquisition, Validation, Investigation, Methodology, Writing – review and editing; Matteo Cantarelli, Conceptualization, Resources, Software, Validation, Investigation, Visualization, Methodology, Writing – review and editing; Robert C Cannon, Conceptualization, Resources, Software, Investigation, Methodology; Andrew P Davison, Software, Investigation, Methodology, Writing – review and editing; Harsha Gurnani, Data curation, Software, Validation, Investigation, Methodology; Robin Angus Silver, Conceptualization, Formal analysis, Supervision, Funding acquisition, Investigation, Methodology, Writing – original draft, Project administration, Writing – review and editing

## Author ORCIDs

Ankur Sinha http://orcid.org/0000-0001-7568-7167
Padraig Gleeson https://orcid.org/0000-0001-5963-8576
Matteo Cantarelli https://orcid.org/0000-0002-0054-226X
Andrew P Davison https://orcid.org/0000-0002-4793-7541
Robin Angus Silver https://orcid.org/0000-0002-5480-6638

Reviewer #1 (Public review): https://doi.org/10.7554/eLife.95135.3.sa1
Reviewer #2 (Public review): https://doi.org/10.7554/eLife.95135.3.sa2
Author response https://doi.org/10.7554/eLife.95135.3.sa3

# Additional files

## Supplementary files

MDAR checklist

## Data availability

No data was generated in this study. All software noted in this manuscript is open source. The NeuroML core libraries can be found at https://github.com/neuroml (copy archived at *Gleeson and Sinha , 2024*). Tables 3 and 4 provide links to the software packages and their source code repositories include DOI information for each software release.

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
