## [Editor Report · eLife Assessment]

This **important** work presents a consolidated overview of the NeuroML2 open community standard and provides **convincing** evidence for its central role within a broader software ecosystem for the development of neuronal models that are open, shareable, reproducible, and interoperable. A major strength of the work is the continued development over more than two decades to establish, maintain, and adapt this standard to meet the evolving needs of the field. This work is of broad interest to the sub-cellular, cellular, computational, and systems neuroscience communities undertaking studies involving theory, modeling, and simulation.

---

## [Referee Report · Reviewer #1 (Public review)]

Summary:

The manuscript gives a broad overview of how to write NeuroML, a brief description of how to use it with different simulators and for different purposes - cells to networks, simulation, optimization and analysis. From this perspective it can be an extremely useful document to introduce new users to NeuroML.

Strengths:

The modularity of NeuroML is indeed a great advantage. For example, the ability to specify the channel file allows different channels to be used with different morphologies without redundancy. The hierarchical nature of NeuroML also is commendable, and well illustrated.

The number of tools available to work with NeuroML is impressive.

Having a python API and providing examples using this API is fantastic. Exporting to NeuroML from python is also a great feature.

The tutorials should assist additional scientists in adopting NeuroML.

Weaknesses:

None noted.

---

## [Referee Report · Reviewer #2 (Public review)]

Summary:

Developing neuronal models that are shareable, reproducible, and interoperable allows the neuroscience community to make better use of published models and to collaborate more effectively. In this manuscript, the authors present a consolidated overview of the NeuroML model description system along with its associated tools and workflows. They describe where different components of this ecosystem lay along the model development pathway and highlight resources, including documentation and tutorials, to help users employ this system.

Strengths:

The manuscript is well-organized and clearly written. It effectively uses the delineated model development life cycle steps, presented in Figure 1, to organize its descriptions of the different components and tools relating to NeuroML. It uses this framework to cover the breadth of the software ecosystem and categorize its various elements. The NeuroML format is clearly described, and the authors outline the different benefits to its particular construction. As primarily a means of describing models, NeuroML also depends on many other software components to be of high utility to computational neuroscientists; these include simulators (ones that both pre-date NeuroML and those developed afterwards), visualization tools, and model databases.

Overall, the rationale for the approach NeuroML has taken is convincing and well-described. The pointers to existing documentation, guides, and the example usages presented within the manuscript are useful starting points for potential new users. This manuscript can also serve to inform potential users of features or aspect of the ecosystem that they may have been unaware of, which could lower obstacles to adoption. While much of what is presented is not new to this manuscript, it still serves as a useful resource for the community looking for information about an established, but perhaps daunting, set of computational tools.

Weaknesses:

The manuscript in large part catalogs the different tools and functionalities that have been produced through the long development cycle of NeuroML. Overall, the interoperability of NeuroML is a benefit, but it does increase the complexity of choices facing users entering into the ecosystem.

In many respects this is an intractable fact of the current environment, but the authors do try to mitigate the issue with user guides (e.g., Table 1) and example code (e.g. Box 1) which address a range of target user audiences, from those learning about the ecosystem for the first time to those looking to implement specific model features. They also categorize different simulator options (Figure 5) and provide feature comparisons (Table 3), which could assist with the most daunting choice faced by new users.

Comments on revised version:

The authors have addressed my major concerns with the original manuscript. The discussion of simulators in particular is much clearer now, and the manuscript has been restructured so that specific details pertinent to a much more focused audience have been rewritten or shifted to more appropriate locations.

---

## [Author Response]

The following is the authors’ response to the original reviews.

**Public Reviews:**

**Reviewer #1 (Public Review):**
Summary:The manuscript gives a broad overview of how to write NeuroML, and a brief description of how to use it with different simulators and for different purposes - cells to networks, simulation, optimization, and analysis. From this perspective, it can be an extremely useful document to introduce new users to NeuroML.

We are glad the reviewer found our manuscript useful.

However, the manuscript itself seems to lose sight of this goal in many places, and instead, the description at times seems to target software developers. For example, there is a long paragraph on the board and user community. The discussion on simulator tools seems more for developers, not users. All the information presented at the level of a developer is likely to be distracting to eLife readership.

To make the paper less developer focussed and more accessible to the end user we have shortened the long paragraphs on the board and user community (and moved some of this text to the Methods section; lines: 524-572 in the document with highlighted changes). We have also made the discussion on simulator tools more focussed on the user (lines 334-406). However, we believe some information on the development and oversight of NeuroML and its community base are relevant to the end user, so we have not removed these completely from the main text.

Strengths:The modularity of NeuroML is indeed a great advantage. For example, the ability to specify the channel file allows different channels to be used with different morphologies without redundancy. The hierarchical nature of NeuroML also is commendable, and well illustrated in Figures 2a through c.The number of tools available to work with NeuroML is impressive.The abstract, beginning, and end of the manuscript present and discuss incorporating NeuroML into research workflows to support FAIR principles.Having a Python API and providing examples using this API is fantastic. Exporting to NeuroML from Python is also a great feature.

We are glad the reviewer appreciated the design of NeuroML and its support for FAIR principles.

Weaknesses:Though modularity is a strength, it is unclear to me why the cell morphology isn't also treated similarly, i.e., specify the morphology of a multi-compartmental model in a separate file, and then allow the cell file to specify not only the files containing channels, but also the file containing the multi-compartmental morphology, and then specify the conductance for different segment groups. Also, after pynml_write_neuroml2_file, you would not have a super long neuroML file for each variation of conductances, since there would be no need to rewrite the multi-compartmental morphology for each conductance variation.

We thank the reviewer for highlighting this shortcoming in NeuroML2. We have now added the ability to reference externally defined (e.g. in another file) and elements from . This has enabled the morphologies and/or specification of ionic conductances to be separated out and enables more streamlined analysis of cells with different properties, as requested. Simulators NEURON, NetPyNE and EDEN already support this new form. Information on this feature has been added to https://docs.neuroml.org/Userdocs/ImportingMorphologyFiles.html#neuroml2 and also mentioned in the text (lines 188-190).

This would be especially important for optimizations, if each trial optimization wrote out the neuroML file, then including the full morphology of a realistic cell would take up excessive disk space, as opposed to just writing out the conductance densities. As long as cell morphology must be included in every cell file, then NeuroML is not sufficiently modular, and the authors should moderate their claim of modularity (line 419) and building blocks (551).

We believe the new functionality outlined above addresses this issue, as a single file containing the element could be referenced, while a much smaller file, containing the channel distributions in a element would be generated and saved on each iteration of the optimisation.

In addition, this is very important for downloading NeuroML-compliant reconstructions from NeuroMorpho.org. If the cell morphology cannot be imported, then the user has to edit the file downloaded from NeuroMorpho.org, and provenance can be lost.

While the NeuroMorpho.Org website does support converting reconstructed morphologies in SWC format to NeuroML, this export feature is no longer supported on most modern browsers due to it being based on Java Applet technologies. However, a desktop version of this application, CVApp, is actively maintained

(https://github.com/NeuroML/Cvapp-NeuroMorpho.org), and we have updated it to support export of the SWC to the standalone element form of NeuroML discussed above. Additionally, a new Python application for conversion of SWC to NeuroML is in development and will be incorporated into PyNeuroML (Google Summer of Code 2024). Our documentation has been updated with the recommended use of SWC in NeuroML based modelling here: https://docs.neuroml.org/Userdocs/Software/Tools/SWC.html

We have also included URLs to the tool and the documentation in the paper (lines: 473-474).

SWC files, however, cannot be used “as is” for modelling since they only include information (often incomplete—for example a single point may represent a soma in SWC files) on the points that make the cell, but not on the sections/segments/cables that these form. Therefore, NeuroML and other simulation tools, including NEURON, must convert these into formats suitable for simulation. The suggested pipeline for use of NeuroMorpho SWC files would therefore be to convert them to NeuroML, check that they represent the intended compartmentalisation of the neuron and then use them in models.

To ensure that provenance is maintained in all NeuroML models (including conversions from other formats), NeuroML supports the addition of RDF annotations using the COMBINE annotation specifications in model files:

https://docs.neuroml.org/Userdocs/Provenance.html. We have added this information to the paper (lines: 464-465).

Also, Figure 2d loses the hierarchical nature by showing ion channels, synapses, and networks as separate main branches of NeuroML.

While an instance of an ion channel is on a segment, in a cell, in a population (and hence there is a hierarchy between them), in terms of layout in a NeuroML file the ion channel is defined at the “top level” so that it can be referenced and used by multiple cells, the cell definitions are also defined top level, and used in multiple populations, etc. There are multiple ways to depict these relationships between entities, and we believe Fig 2d complements Fig 2a-c (which is more hierarchical), by emphasising the different categories of entities present in NeuroML files. We have modified the caption of Figure 2d to clarify that it shows the main categories of elements included in the NeuroML standard in their respective hierarchies.

In Figure 5, the difference between the core and native simulator is unclear.

We have modified the figure and text (lines: 341) to clarify this. We now say “reference” simulators instead of “core”. This emphasises that jNeuroML and pyLEMS are intended as reference implementations in each of their languages of how to interpret NeuroML models, as opposed to high performance simulators for research use. We have also updated the categorization of the backends in the text accordingly.

What is involved in helper scripts?

Simulators such as NetPyNE can import NeuroML into their own internal format, but require some boilerplate code to do this (e.g. the NetPyNE scripts calls the importNeuroML2SimulateAnalyze() method with appropriate parameters). The NeuroML tools generate short scripts that use this boilerplate code. We have renamed “helper scripts” to “import scripts'' for clarity (Figure 5 and its caption).

I thought neurons could read NeuroML? If so, why do you need the export simulator-specific scripts?

The NEURON simulator does have some NeuroML functionality (it can export cells, though not the full network, to NeuroML 2 through its ModelView menu), but does not natively support reading/importing of NeuroML in its current version. But this is not a problem as jNeuroML/PyNeuroML translates the NeuroML model description into NEURON’s formats: Python scripts/HOC/Nmodl which NEURON then executes.

As NEURON is the simulator which allows simulation of the widest range of NeuroML elements, we have (in agreement with the NEURON developers) concentrated on incorporating the best support for NeuroML import/export in the latest (easy to install/update) releases of PyNeuroML, rather than adding this to the Neuron source code. NEURON’s core features have been very stable for years and many versions of the simulator are used by modellers - installing the latest PyNeuroML gives them the latest NEURON support without having to reinstall the latter.

In addition, it seems strange to call something the "core" simulation engine, when it cannot support multi-compartmental models. It is unclear why "other simulators" that natively support NeuroML cannot be called the core.

We agree that this terminology was confusing. As mentioned above, we have changed “core simulator” to “reference simulator”, to emphasise the roles of these simulation engine options.

It might be more helpful to replace this sort of classification with a user-targeted description. The authors already state which simulators support NeuroML and which ones need code to be exported. In contrast, lines 369-370 mention that not all NeuroML models are supported by each simulator. I recommend expanding this to explain which features are supported in each simulator. Then, the unhelpful separation between core and native could be eliminated.

As suggested, we have grouped the simulators in terms of function and removed the core/ non-core distinction. We have also added a table (Table 3) in the appendices that lists what features each simulation engine supports and updated the text to be more user focussed (lines: 348-394).

The body of the manuscript has so much other detail that I lose sight of how NeuroML supports FAIR. It is also unclear who is the intended audience. When I get to lines 336-344, it seems that this description is too much detail for the eLife audience. The paragraph beginning on line 691 is a great example of being unclear about who is the audience. Does someone wanting to develop NeuroML models need to understand XSD schema? If so, the explanation is not clear. XSD schema is not defined and instead explains NeuroML-specific aspects of XSD. Lines 734-735 are another example of explaining to code developers (not model developers).

We have modified these sentences to be more suitable for the general eLife audience: we have moved the explanation of how the different simulator backends are supported to the more technically detailed Methods section (lines 882-942).

While the results sections focus on documenting what users can do with NeuroML, the Methods sections include information on “how” the NeuroML and software ecosystem function. While the information in the methods sections may not be required by users who want to use the standard NeuroML model elements, those users looking to extend NeuroML with their own model entities and/or contribute these for inclusion in the NeuroML standard will require some understanding of how the schema and component types work.

We have tried to limit this information to the bare minimum, pointing to online documentation where appropriate. XSD schemas are, for example, briefly introduced at the beginning of the section “The NeuroML XML Schema”. We have also included a link to the W3C documentation on XSD schemas as a footnote (line 724).

**Reviewer #2 (Public Review):**
Summary:Developing neuronal models that are shareable, reproducible, and interoperable allows the neuroscience community to make better use of published models and to collaborate more effectively. In this manuscript, the authors present a consolidated overview of the NeuroML model description system along with its associated tools and workflows. They describe where different components of this ecosystem lay along the model development pathway and highlight resources, including documentation and tutorials, to help users employ this system.Strengths:The manuscript is well-organized and clearly written. It effectively uses the delineated model development life cycle steps, presented in Figure 1, to organize its descriptions of the different components and tools relating to NeuroML. It uses this framework to cover the breadth of the software ecosystem and categorize its various elements. The NeuroML format is clearly described, and the authors outline the different benefits of its particular construction. As primarily a means of describing models, NeuroML also depends on many other software components to be of high utility to computational neuroscientists; these include simulators (ones that both pre-date NeuroML and those developed afterwards), visualization tools, and model databases.Overall, the rationale for the approach NeuroML has taken is convincing and well-described. The pointers to existing documentation, guides, and the example usages presented within the manuscript are useful starting points for potential new users. This manuscript can also serve to inform potential users of features or aspects of the ecosystem that they may have been unaware of, which could lower obstacles to adoption. While much of what is presented is not new to this manuscript, it still serves as a useful resource for the community looking for information about an established, but perhaps daunting, set of computational tools.

We are glad the reviewer appreciated the utility of the manuscript.

Weaknesses:The manuscript in large part catalogs the different tools and functionalities that have been produced through the long development cycle of NeuroML. As discussed above, this is quite useful, but it can still be somewhat overwhelming for a potential new user of these tools. There are new user guides (e.g., Table 1) and example code (e.g. Box 1), but it is not clear if those resources employ elements of the ecosystem chosen primarily for their didactic advantages, rather than general-purpose utility. I feel like the manuscript would be strengthened by the addition of clearer recommendations for users (or a range of recommendations for users in different scenarios).

To make Table 1 more accessible to users and provide recommendations we have added the following new categories: Introductory guides aimed at teaching the fundamental

NeuroML concepts; Advanced guides illustrating specific modelling workflows; and Walkthrough guides discussing the steps required for converting models to NeuroML. Box 1 has also been improved to clearly mark API and command line examples.

For example, is the intention that most users should primarily use the core NeuroML tools and expand into the wider ecosystem only under particular circumstances? What are the criteria to keep in mind when making that decision to use alternative tools (scale/complexity of model, prior familiarity with other tools, etc.)? The place where it seems most ambiguous is in the choice of simulator (in part because there seem to be the most options there) - are there particular scenarios where the authors may recommend using simulators other than the core jNeuroML software?The interoperability of NeuroML is a major strength, but it does increase the complexity of choices facing users entering into the ecosystem. Some clearer guidance in this manuscript could enable computational neuroscientists with particular goals in mind to make better strategic decisions about which tools to employ at the outset of their work.

As mentioned in the response to Reviewer 1, the term “core simulator” for jNeuroML was confusing, as it suggested that this is a recommended simulation tool. We have changed the description of jNeuroML to a “reference simulator” to clarify this (Figure 5 and lines 341, 353).

In terms of giving specific guidance on which simulator to use, we have focussed on their functionality and limitations rather than recommending a specific tool (as simulator independent standards developers we are not in a position to favour particular simulators). While NEURON is the most widely used simulator currently, other simulation opinions (e.g. EDEN) have emerged recently which provide quite comprehensive NeuroML support and similar performance. Our approach is to document and promote all supported tools, while encouraging innovation and new developments. The new Table 3 in the Appendix gives a guide to assist users in choosing which simulator may best suit their needs and we have updated the text to include a brief description (lines 348-394).

**Recommendations for the authors:**

**Reviewer #1 (Recommendations For The Authors):**
I do not understand what the $comments mean in Box 1. It isn't until I get further in the text that I realize that those are command line equivalents to the Python commands.

We thank the reviewer for highlighting this confusion. We’ve now explicitly marked the API usage and command line usage example columns to make this clearer. We have also used “>” instead of “$” now to indicate the command line,

In Figure 9 Caption "Examples of analysis functions ..", the word analysis seems a misnomer, as these graphs all illustrate the simulation output and graphing of existing variables. I think analysis typically refers to the transformation of variables, such as spike counts and widths.

To clarify this we have changed the caption to “Examples of visualizing biophysical properties of a NeuroML model neuron”.

Figure 10: Why is the pulse generator part of a model? Isn't that the input to a model?

Whether the input to the model is described separately from the NeuroML biophysical description or combined with it is a choice for the researcher. This is possible because in NeuroML any entity which has time varying states can be a NeuroML element, including the current pulse generator. In this simple example the input is contained within the same file (and therefore element) as the cell. However, this does not need to be the case. The cell could be fully specified in its own NeuroML file and then this can be included in other files which add different inputs to facilitate different simulation scenarios. The Python scripting interface facilitates these types of workflows.

In the interest of modularity, can stim information be stored in a separate file and "included"?

Yes, as mentioned above, the stimulus could be stored in a separate file.

I find it strange to use a cell with mostly dimensionless numbers as an example. I think it would be more helpful to use a model that was more physiological.

In choosing an example model type to use to illustrate the use of LEMS (Fig 12), NeuroML (Fig 10), XML Schema (Fig 11), the Python API (Fig 13) and online documentation (Fig 15), we needed an example which showed a sufficiently broad range of concepts (dimensional parameters, state variables, time derivatives), but which is sufficiently compact to allow a concise depiction of the key elements in figures, that fit in a single page (e.g. Fig 12). We felt that the Hindmarsh Rose model, while not very physiological, was well suited for this purpose (explaining the underlying technologies behind the NeuroML specification). The simplicity of the Hindmarsh Rose model is counterbalanced in the manuscript by the detailed models of neurons and circuits in Figures 7 & 9. The latter shows a morphologically and biophysically detailed cortical L5b pyramidal cell model.

In lines 710-714, it is unclear what is being validated. That all parameters are defined? Using the units (or lack thereof) defined in the schema?

Validation against the schema is “level 1” validation where the model structure, parameters, parameter values and their units, cardinality, and element positioning in the model hierarchy are checked. We have updated the paragraph to include this information and to also point to Figure 6 where different levels of validation are explained.

Lines 740 to 746 are confusing. If 1-1 between XSD and LEMS (1st sentence) then how can component types be defined in LEMS and NOT added to the standard? Which is it? 1-1 or not 1-1?

For the curated model elements included in the NeuroML standard, there will be a 1-1 correspondence between their component type definitions in LEMS and type definitions in the XSD schema. New user defined component types (e.g. a new abstract cell model) can be specified in LEMS as required, and these do not need to be included in the XSD schema to be loaded/simulated. However, since they are not present in the schema definition of the core/curated elements, they cannot be validated against it (level 1 validation). We have modified the text to make this clearer (line: 778).

Nonetheless, if the new type is useful for the wider community, it can be accepted by the Editorial Board, and at that stage it will be incorporated into the core types, and added to the Schema, to be part of “valid NeuroML”.

Figure 12. select="synapses[*]/i" is not explained. Does /i mean that iSyn is divided by i, which is current (according to the sentence 3 lines after 766) or perhaps synapse number?

We thank the reviewer for highlighting this confusion. We have now explained the construct in the text (lines 810-812). It denotes “select the i (current) values from all Attachments which have the id ‘synapses’”. These multiple values should be reduced down to a single value through addition, as specified by the attribute: reduce=”add”.

The line after 766 says that "DerivedVariables, variables whose values depend on other variables". You should add "and that are not derivatives, which are handled separately" because by your definition derivatives are derived variables.

Thank you. We have updated the text with your suggestion

**Reviewer #2 (Recommendations For The Authors):**
- Figure 9: I found it somewhat confusing to have the header from the screenshot at the top ("Layer 5 Burst Accommodating Double Bouquet Cell (5)") not match the morphology shown at the bottom. It's not visually clear that the different panels in Figure 9 may refer to unrelated cells/models.

Thank you for pointing this out. We have replaced the NeuroML-DB screenshot with one of the same Layer 5b pyramidal cells shown in the panels below it.

Additional change:

Figure 7c (showing the NetPyNE-UI interface) has been replaced. Previously, this displayed a 3D model which had been created in NetPyNE itself, but now shows a model which has been created in NeuroML and imported for display/simulation in NetPyNE-UI, and therefore better illustrates NeuroML functionality.